# ONLINE POLICY SELECTION FOR INVENTORY PROBLEMS

## ABSTRACT

We tackle online inventory problems where at each time period the manager makes a replenishment decision based on partial historical information in order to meet demands and minimize costs. To solve such problems, we build upon recent works in online learning and control, use insights from inventory theory and propose a new algorithm called GAPSI. This algorithm follows a new feature-enhanced base-stock policy and deals with the troublesome question of non-differentiability which occurs in inventory problems. Our method is illustrated in the context of a complex and novel inventory system involving multiple products, lost sales, perishability, warehouse-capacity constraints and lead times. Extensive numerical simulations are conducted to demonstrate the good performances of our algorithm on real-world data.

## 1 INTRODUCTION

Inventory control is a standard problem in operations research and operations management where a manager needs to make replenishment decisions in order to minimize costs and meet demands (Nahmias, 2011; Roldán et al., 2017). Classical inventory theories focus on the optimization side of the problem, that is, determining what is the optimal replenishment strategy assuming all the parameters of the inventory model are known. For instance, the manager knows future demands or the distribution they are drawn from. However, in real-world scenarios an inventory manager rather faces optimization and learning problems presented in a sequential fashion, that is, at each time period the manager makes its replenishment decisions based on past observations. For these reasons, machine learning frameworks such as online learning or reinforcement learning seem particularly well-suited to solve these realistic inventory problems. For instance, there is now an important body of literature that builds upon techniques from online convex optimization like online gradient descent (Zinkevich, 2003) to solve inventory problems (Hihat et al., 2024). However, they are extremely specialized, designed for specific inventory models involving specific dynamics, cost structures and demand processes (see, for example, Huh & Rusmevichientong, 2009; Shi et al., 2016; Zhang et al., 2018). On the other hand, other generic approaches for control problems such as Model Predictive Control (MPC) (Mattingley et al., 2011) suffer from a prohibitive computational cost and a lack of scalability.

**Contributions.** In this paper, we first show how realistic, general inventory problems fit within the recent Online Policy Selection (OPS) framework of Lin et al. (2024) which is at the crossroads of control and online learning. We detail how various constraints very common in the industry, such as perishability, lead times, or warehouse capacity constraints, can be mathematically modeled in this setting. We obtain a general online inventory problem, that can be simplified and applied to several more classical problems such as usual perishable inventory systems (Nahmias, 2011).

We then present a new algorithm for solving this online optimization problem, called GAPSI. This algorithm is adapted from GAPS (Lin et al., 2024) to take into account specific aspects of inventory problems, in particular the fact that the functions we are dealing with are not differentiable. We find that this non-differentiability problem cannot be ignored, since it leads to undesirable behaviors, and we show how carefully selected derivatives for the policies and Adagrad-style learning rates solves it. We also propose a new policy which draws on classical base-stock policies (Snyder & Shen, 2019, Section 4.3.1) while dealing with uncertainty in future demand. The idea is to learn a target

level which writes as a linear function of either past demand features or forecasts. In this way, we obtain a general online method for solving real-world inventory problems that can take into account many different constraints and domain-specific knowledge about demands. For example, if we know that demand has a weekly seasonality, then this knowledge can be incorporated into the algorithm by using indicators of the day of the week as features.

Finally, we provide extensive experiments which demonstrate the good performance of GAPSI compared to classical approaches. We observe that GAPSI performs particularly well when demands are not stationary, have some changes of regime, and when the features are well-chosen. We emphasise that in this work we do not only propose an efficient new algorithm, but also want to show that online learning is a promising approach for realistic inventory problems, and therefore draw the attention of this community to these problems.

**Related works.** Many online learning frameworks have been applied or adapted to inventory problems. For instance, in the absence of inventory dynamics, Levina et al. (2010) see their problem as prediction with expert advice and Lugosi et al. (2024) use partial monitoring. In the presence of dynamics, the situation is much more complex, and this is what interests us. The research most closely related to this paper typically considers Online Convex Optimization (OCO) (Zinkevich, 2003) as their main learning framework. This is the case of Huh & Rusmevichientong (2009), Shi et al. (2016), Zhang et al. (2018), Zhang et al. (2020), Guo et al. (2024). The common approach in this literature is to adapt OCO techniques, such as Online Gradient Descent (OGD), in various ways to handle inventory dynamics. However, this is mostly done on a case-by-case basis: specialized algorithms are designed for specific dynamics. So the manager can only implement these solutions if they are facing a very similar problem.

The reason why most inventory problems are not instances of classical online learning problems is that they lack a notion central in control problems: that of a dynamical system influenced by the manager's decisions and impacting the losses. Incorporating this notion in online learning leads us to the literature of online control. The latter has mostly focused on linear dynamics, as in Agarwal et al. (2019), but most inventory dynamics are not linear. Among the recent developments in online control, the OPS framework of Lin et al. (2024), presented in Subsection 2.1, provides much more flexibility by allowing for general dynamics. Nevertheless, some challenges remain when considering inventory problems through OPS among which the non-differentiability of the losses, policies and dynamics.

On the other hand, there exist general-purpose control techniques which are not specific to inventory problems, in particular, Model Predictive Control (MPC) (Mattingley et al., 2011), which became popular in the 1980s. The main idea is to solve an optimization problem at each time step based on a predictive model up to a receding planning horizon. These approaches are not the main subject of this paper, but we consider MPC as a competitor in the experimental section.

**Overview.** We start in Section 2 by introducing the OPS framework and show through various examples that it is well-adapted to model realistic inventory problems. Then, we present in Section 3 our algorithm, GAPSI, that is based on GAPS (Lin et al., 2024) while taking into account aspects specific to inventory management, via the use of a feature-enhanced base-stock policy, AdaGrad learning rates and carefully chosen partial derivatives. We conclude in Section 4 with an extensive numerical simulation study. We refer to the appendix for precise mathematical details and additional experiments.

**Notations.** Let us denote by $\mathbb{R}_+ = [0, +\infty)$ the set of non-negative real numbers, $\mathbb{N} = \{1, 2, \dots\}$ the set of positive integers, $\mathbb{N}_0 = \{0, 1, \dots\}$ the set of non-negative integers and $[n] = \{1, \dots, n\}$.

## 2 PROBLEM STATEMENT

### 2.1 ONLINE POLICY SELECTION

The Online Policy Selection (OPS) framework of Lin et al. (2024) is a discrete-time control problem where the decision-maker learns the parameters of a parameterized policy in an online fashion. Formally, let $\mathbb{X}$ be the state space, $\mathbb{U}$ the control space and $\Theta$ the parameter space. At each time

period $t \in \mathbb{N}$, the decision-maker starts by observing the state of the system $x_t \in \mathbb{X}$, then, they choose a parameter $\theta_t \in \Theta$ which is used to determine the control through a time-varying policy: $u_t = \pi_t(x_t, \theta_t) \in \mathbb{U}$. A loss $\ell_t(x_t, u_t) \in \mathbb{R}$ is incurred, and finally the system transitions to the next state: $x_{t+1} = f_t(x_t, u_t) \in \mathbb{X}$. Given a horizon $T \in \mathbb{N}$, the goal is to minimize the cumulative loss $\sum_{t=1}^{T} \ell_t(x_t, u_t)$ by selecting $\theta_t$ sequentially. We assume that the dynamics $(f_t)_{t \in \mathbb{N}}$, losses $(\ell_t)_{t \in \mathbb{N}}$, and policies $(\pi_t)_{t \in \mathbb{N}}$ are oblivious, meaning they are fixed before the interaction starts.

Note that the well-studied Online Convex Optimization (OCO) framework (Zinkevich, 2003) can be seen as a special case of OPS, obtained by removing the dynamics of the system, simplifying the policies, and introducing convexity assumptions. Formally, to recover OCO, one can set $\mathbb{X} = \{0\}$ and $\pi_t(x_t, \theta_t) = \theta_t$, assume that $\Theta$ is a closed convex subset of an Euclidean space and that $u_t \mapsto \ell_t(x_t, u_t)$ is convex for every $t \in \mathbb{N}$.

## 2.2 INVENTORY PROBLEMS

To model an inventory, we must determine its dynamics and the cost structure to be minimized. The vector $x_t$ must fully describe the state of the inventory at time $t$. It encodes the quantities of products, including both on-hand units and eventual on-order units. The control $u_t$ is used here to represent the ordered quantities at time $t$. The transitions $f_t$ will determine how the inventory states evolve over time. Loss functions $\ell_t$ determine the cost structure. We describe below how these quantities can be defined, starting with a simple model and making it more complex as we go along.

**Lost sales.** First, assume that we have one product and no lead time, meaning that a product is instantaneously received when ordered and no perishability. Then, $x_t \in \mathbb{R}_+$ is simply the quantity of this product available in the inventory. We assume that unmet demand is lost, which is modeled by the transition $f_t(x_t, u_t) = [x_t + u_t - d_t]^+$, where $d_t \in \mathbb{R}_+$ is the demand at time period $t$. Online lost sales inventory problems have been investigated by Huh & Rusmevichientong (2009) for instance.

**Fixed lifetime perishability.** Now, if this product has a fixed usable lifetime of $m \in \mathbb{N}$ periods, then a unit received on period $t$ can be sold from periods $t$ to $t + m - 1$ and, if this does not happen, it expires and leaves the inventory at the end of period $t + m - 1$ as an outdating unit. To model such a system we need to keep track of the entire age distribution of the on-hand inventory through a state vector $x_t$ of dimension $n = m - 1$ (Nahmias, 2011, Section 1.3). For $i \in [m - 1]$, the $i^{th}$ coordinate $x_{t,i}$ represent the number of units that will expire at the end of day $t + i - 1$ if not sold before. We again assume that unmet demand is lost and we choose a classical transition $f_t$ which sells first the oldest products (Nahmias, 2011, Chapter 2). This can be written component-wise as

$$f_{t,i}(x_t, u_t) = f_{t,i}(z_t) = \left[ z_{t,i+1} - \left[ d_t - \sum_{j=1}^{i} z_{t,i} \right]^+ \right]^+, \tag{1}$$

where $z_t = (x_{t,1}, \ldots, x_{t,n}, u_t) \in \mathbb{R}^{n+1}$ will be thereafter called the state-control couple. Zhang et al. (2018) proposed an online control algorithm for such systems.

**Order lead times.** An order lead time, also known as order delay, is the difference between the reception time period and the order time period. To adapt our model to take into account order lead times on top of tracking on-hand products, we need to track on-order products in the state vector by increasing its dimension. For instance, to include a lead time $L \in \mathbb{N}_0$ in lost sales perishable inventory systems, we can set $n = m + L - 1$. Then, the first $m - 1$ coordinates of the state $x_t$ evolve following (1) and the next $L$ coordinates correspond to on-order units. To our knowledge there exist online algorithms handling lead times (Zhang et al., 2020; Agrawal & Jia, 2022), but they consider only non-perishable products.

**Multi-product system.** Assume now that there are $K \in \mathbb{N}$ product types indexed by $k \in [K]$. Each product is perishable with lifetime $m_k \in \mathbb{N}$ and has a lead time $L_k \in \mathbb{N}_0$. We add an index $k$ to all quantities which are product-dependent. For example, we now have $K$ transition functions $(f_{t,k}(x_t, u_t))_{k \in [K]}$, $K$ demands per time step $(d_{t,k})_{k \in [K]}$, and the state-control couples are denoted by $z_t = (z_{t,k})_{k \in [K]}$ where each $z_{t,k} = (x_{t,k,1}, \ldots, x_{t,k,n_k}, u_{t,k})$. Multi-product systems

become interesting whenever losses or transitions cannot be separated per product. This happen in the presence of joint constraints like the warehouse-capacity constraints introduced next.

**Warehouse-capacity constraints.**  Until now, we have assumed that one can store an arbitrarily large quantity of products in the inventory, but in real-world problems we may need to ensure that the warehouse's capacity is not overflown. For instance, Shi et al. (2016) consider this kind of constraints in an online control setting, but with no lead times nor perishability. Modeling warehouse-capacity constraints can be nontrivial, in particular in presence of lead times, since the manager does not know the future demands in advance. We propose a new model imposing restrictions at reception time to prevent overflow, which we summarize next and whose details can be found in Appendix A. Given a state-control vector $z_t$, we check whether a warehouse-capacity constraint $z_t \in \mathbb{V}_t$ is satisfied or not. If the constraint is not satisfied, some operator must be applied to $z_t$ to discard some units. We propose to remove products that just arrived in the ascending order (starting from product $k = 1, 2, \ldots, K$), which yields a new state-control vector $\tilde{z}_{t,k}$. This allows us to define new transitions $f_t$, by taking any of the previously seen transitions and evaluating it at $\tilde{z}_t$ instead of $z_t$.

**Losses.**  The goal in inventory control can be seen as minimizing a loss $\ell_t$ writing as a sum of terms capturing different trade-offs such as over-ordering versus under-ordering, or meeting the demand while maintaining low inventory management costs. For instance, the *penalty cost* is a term proportional to unmet demand, which writes $[d_{t,k} - \sum_{i=1}^{m_k} \tilde{z}_{t,k,i}]^+$. Similarly, the *holding cost* is a term proportional to on-hand units just after meeting demand, $[\sum_{i=1}^{m_k} \tilde{z}_{t,k,i} - d_{t,k}]^+$. These are the most classical costs for losses in stochastic inventory problems, see, e.g., the newsvendor problem in Arrow et al. (1951) or Snyder & Shen (2019, Subsection 3.1.3). We could also incorporate usual terms such as the *purchase cost* (proportional to ordered units) or the *outdating cost* (is proportional to outdating units). We also propose a new *overflow cost*, proportional to discarded units due to overflow, which is specific to our model and pushes the manager to respect the warehouse-capacity constraint. In what follows, we will assume that $\ell_t$ is the sum of these five costs (see Appendix A for details).

**Summary.**  Putting everything together, the following timeline summarizes our model in its most general form. For each time period $t = 1, 2, \ldots$

1. The manager observes the state $x_t \in \mathbb{X}$ (which is zero if $t = 1$).

2. The manager orders the quantities $u_t \in \mathbb{U}$ and pays purchase costs.

3. The manager receives the units $(z_{t,k,m_k})_{k \in [K]}$, some of which may be discarded due to the warehouse-capacity constraint, incurring overflow costs and forming a new state-control vector $\tilde{z}_t$.

4. The demand $d_t$ is realized and met to the maximum extent possible using on-hand units $(\sum_{i=1}^{m_k} \tilde{z}_{t,k,i})_{k \in [K]}$ starting by oldest units $(\tilde{z}_{t,k,1})_{k \in [K]}$. Penalty costs and holding costs are paid.

5. The next inventory state $x_{t+1} \in \mathbb{X}$ is defined through the transition $f_t$, where outdating units leave the inventory incurring an outdating cost.

## 3 THE ALGORITHM: GAPSI

Let us now introduce our new algorithm for inventory problems named Gradient-based Adaptive Policy Selection for Inventories (GAPSI). Its pseudo-code is provided in Algorithm 1. In essence, GAPSI performs an online gradient descent to update the parameters of its replenishment policy. More precisely, GAPSI follows a new feature-enhanced base-stock policy whose parameter is sequentially updated according to AdaGrad (Streeter & McMahan, 2010; Duchi et al., 2011), using an approximated gradient computed as in GAPS (Lin et al., 2024) by combining carefully chosen generalized Jacobian matrices. We detail further what each step does below.

### 3.1 DESIGNING A POLICY TAILORED TO INVENTORY PROBLEMS

The first action the decision-maker needs to perform is to pass an order based on the current state $x_t$, according to a certain policy $\pi_t$, which usually depends on $x_t$ and some parameter. A standard

---

**Algorithm 1:** GAPSI

1 **Parameters:** Learning rate factor $\eta > 0$, buffer size $B \in \mathbb{N}$, initial parameter $\theta_1 \in \Theta$.
2 Observe the initial state $x_1$;
3 **for** $t = 1, 2, \ldots$ **do**
4  Order the quantities $u_t = \pi_t(x_t, \theta_t)$ according to the policy $\pi_t$ (see Section 3.1) ;
5  Incur the loss $\ell_t(x_t, u_t)$ and observe the next state $x_{t+1} = f_t(x_t, u_t)$;
6  Compute the Jacobian matrices of $\pi_t$, $\ell_t$ and $f_t$ (see Section 3.3);
7  Compute an approximated gradient $g_t$ as in GAPS (see Section 3.2);
8  Update the parameter $\theta_{t+1}$ by using $g_t$ via AdaGrad (see Section 3.2);

---

policy for stochastic inventory problems is the *base-stock* policy (Snyder & Shen, 2019, Chapter 4). Such policy is parameterized by a time-varying base-stock level $S_t \in \mathbb{R}_+^K$ which the manager tries to maintain: for each product, if the inventory position is less than the base-stock level then the difference is ordered, otherwise no order is placed. This policy is appealing because it is optimal in a certain sense for simple problems, see Snyder & Shen (2019, Section 4.5), Bu et al. (2023), and Xie et al. (2024).

For GAPSI, we introduce a *feature-enhanced* variant of this policy. We assume that before each order at time $t$, and for every product $k \in [K]$, the manager has access to a vector of nonnegative features $w_{t,k} \in \mathbb{R}_+^{p_k}$. These can typically gather information about the seasonality, holidays, price discounts or demand forecasts. We then propose to follow a base-stock policy whose level $S_{t,k}$ is, for each product $k$, a linear combination of the features $w_{t,k,i}$ with some coefficients $\theta_{t,k,i}$ which we need to learn. Our choice of policy can then be formally defined as:

$$\pi_t(x_t, \theta_t) = (\pi_{t,k}(x_{t,k}, \theta_{t,k}))_{k \in [K]} \quad \text{where} \quad \pi_{t,k}(x_{t,k}, \theta_{t,k}) = \left[ w_{t,k}^\top \theta_{t,k} - \sum_{i=1}^{m_k + L_k - 1} x_{t,k,i} \right]^+.$$

Note that if we take univariate constant feature vectors (such as $w_{t,k} \equiv 1$) we recover standard base-stock policies. We also point out that if the features are forecasts of the demand, our policy recovers an heuristic proposed by Motamedi et al. (2024, Subsection 5.1) in the context of single-product offline inventory problems. Finally, we highlight that $\pi_t$ is not differentiable, which happens to be a problem when optimizing with respect to $\theta$, which will be discussed in Section 3.3.

### 3.2 LEARNING THE PARAMETERS WITH GAPS AND ADAGRAD

The main goal of GAPSI is to learn the parameters $\theta_t \in \Theta$, which we assume to be constrained in a box $\Theta = \prod_{i=1}^P [a_i, b_i]$, $P := \sum_k p_k$. To do so, we use an online optimization scheme which approximately minimizes a surrogate loss function $L_t : \Theta \to \mathbb{R}$. Precisely, $L_t(\theta)$ is the loss which we would have incurred at time $t$ if we had followed the policy associated to $\theta$ *for all periods so far*, that is, if we had applied the controls $u_s = \pi_s(x_s, \theta)$ for all $s = 1, \ldots, t$.

Ideally, and this is a standard idea in online control, we would like to perform an online gradient descent with respect to $L_t$, but the cost of computing $\nabla L_t(\theta_t)$ is prohibitive when $t$ grows. This is why we turn to GAPS (Lin et al., 2024), a procedure returning an approximated gradient $g_t \sim \nabla L_t(\theta_t)$ at a reasonable cost, by making two approximations. First, instead of computing $\nabla L_t(\theta_t)$ along the ideal trajectory of parameters $(\theta_t, \ldots, \theta_t)$, it is computed along the current trajectory $(\theta_1, \ldots, \theta_t)$, allowing to use efficiently past computations. Second, the historical dependency is truncated to the $B$ most recent time steps. By doing so, all we need to do at each time step is to compute the jacobians of the functions $\pi_t, f_t, \ell_t$ and to combine them with past jacobians to calculate $g_t$?. For more details on the implementation, we refer to Lin et al. (2024) and Appendix B.

Once we have computed the approximated gradient $g_t$, GAPSI can update the parameter $\theta_t$ by performing one step of AdaGrad (Streeter & McMahan, 2010; Duchi et al., 2011), where the learning rates are set component-wise as in Orabona (2019, Algorithm 4.1) and can be further tuned with an extra parameter $\eta > 0$:

$$\theta_{t+1} = \text{Proj}_\Theta (\theta_t - H_t g_t) \quad \text{with} \quad H_t = \text{diag}(\eta_{t,1}, \ldots, \eta_{t,P}) \quad \text{and} \quad \eta_{t,i} = \eta \frac{b_i - a_i}{\sqrt{\sum_{s=1}^t g_{s,i}^2}}. \quad (2)$$

Our choice of this variant of AdaGrad is motivated by its adaptivity to the gradients, its coordinate-wise learning process and its decreasing learning rates.

### 3.3 THE TROUBLESOME COMPUTATION OF JACOBIANS FOR NONSMOOTH FUNCTIONS

As described in Section 3.2, at each iteration we need to compute the jacobians[1] of the functions $\pi_t$, $f_t$, and $\ell_t$. A striking feature of these functions is that none of them is differentiable, due to the presence of positive parts in their definition (see Sections 2.2 and 3.1). In standard machine learning, non-differentiability may cause theoretical difficulties (Bolte & Pauwels, 2021), but in practice it is usually not a problem: most neural network architectures include ReLU (the positive part function) but still perform perfectly well. This apparent contradiction can be ignored by observing that, in general, points of non-differentiability are never reached during training (Bertoin et al., 2021, Theorem 2).

However, this story appears to be surprisingly different for online inventory problems. First, different choices of subgradients can lead to drastically different trajectories for GAPSI, and some of them can lead to disastrous performance. Second, in some real-world scenarios most jacobians simply cannot be accessed. Therefore, the main message of this section is that one cannot blindly rely on automatic differentiation for such nonsmooth online problems. We explain below where those problems come from, and how to avoid them.

**Differentiating the policy $\pi_t$.** Imagine a scenario in which the demand for a product is zero on a given interval of time, pushing the manager to reduce the corresponding stock to zero. Then arises the question of what happens when the demand becomes positive again. One would expect that the manager starts to order again the said product. But it appears that this depends heavily on how the partial derivatives of $\pi_t$ are computed. To see this, look at Figure 1 where we simulate a simple problem where the demand is 0 for 100 days, and then switches to 1 for the next 100 days. When running GAPSI with standard autodifferentiation rules for derivating $\pi_t$, one can see that after the 100th day the base-stock level remains stationary: the manager keeps the inventory level to 0, missing numerous sales. A simple analysis (see Appendix B.2 for the details) shows that because autodiff computes the left-partial derivative of $\pi_t$, as soon as $\theta_t = 0$ the parameter will remain this way even if the demand restarts. Therefore, we advocate for always taking the *right*-partial derivatives of $\pi_t$. Our custom differentiation rule can be seen in action in Figure 1. Whenever the level reaches zero, our differentiation rule implies that a negative gradient is computed, therefore increasing the level. This leads to oscillations which decay thanks to the Adagrad learning rate (2). On the other hand, as soon as the level reaches zero, auto-differentiation leads to a zero gradient, independently of the demand which leads to this undesirable stationary behavior. We highlight that zero demand for a long time interval is not specific to the above toy problem, but can often be observed in real-world problems (see Figure 4 for example).

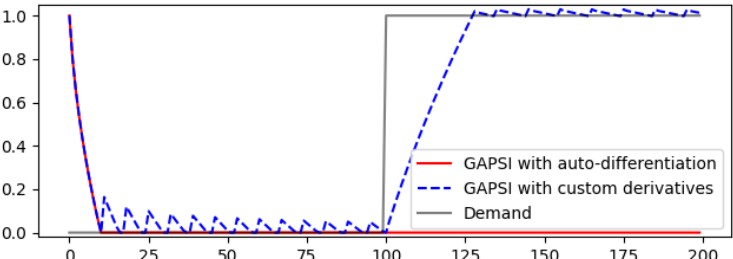

Figure 1: Base-stock level w.r.t. time when using GAPSI, depending on the differentiation rule.

**Differentiating the loss $\ell_t$ and transition $f_t$.** The loss $\ell_t$ depends on the parameters at hand, structural parameters of the problem (such as unit costs or volume information), but also on the exogenous demand $d_t$. It is a well-known issue in inventory problems that the demand $d_t$ is sometimes unknown even at the end of time $t$, preventing the computation of the loss $\ell_t$ or its derivatives, called

---

[1]Because we consider nonsmooth functions, it would be more rigorous to talk about *generalized* jacobians but for the sake of simplicity we will simply call them jacobians.

the censored demand framework. However, in this framework it is usually accepted that the manager has access to a partial information, which is the number of sales. Previous work such as Huh & Rusmevichientong (2009) observed in such a framework that even if the subdifferential of $\ell_t$ cannot be accessed, its *left*-partial derivatives can still be computed. This observation can be adapted to our model, at least when there are no warehouse-capacity constraints. The situation is actually exactly the same for the transition $f_t$. We observe that its subdifferential cannot be accessed in general, but that its left-partial derivatives can be computed, which to our knowledge has never been discussed in prior works.

All derivations of these partial derivatives are given in Appendix B and implemented in the code. As a side benefit, we get a faster algorithm with these custom derivatives than if we had used autodifferentiation directly.

## 4 NUMERICAL EXPERIMENTS

In this section we evaluate empirically the performances of GAPSI. We refer to Appendix C for additional details and results.

**Datasets.** In this section, we use two real-world datasets, the M5 dataset provided by the company Walmart (Makridakis et al., 2022), and a proprietary dataset from the company Califrais, a food supply chain start-up. Both datasets include multiple sales time series over horizons of $T = 1969$ and $T = 860$ days respectively. The time series are organized hierarchically: the total demand is split into categories which are split into subcategories, which are finally split into products. These datasets are therefore very rich depending on where we place ourselves in the hierarchy, the total demand (root of the hierarchy) having much less variability than the demand of one specific product (leaves of the hierarchy). We will use different levels in this hierarchy, treating the time series as demand for a single product, even if it is actually aggregated over different products (of a category or all of them in the case of the total demand).

**Metrics.** To compare algorithms, our metric throughout the section is the ratio of cumulative losses between the considered algorithm and the best stationary base-stock policy $S_T^*$. This policy picks the single best base-stock level independently of time, given the demand realizations over a horizon $T$. It is therefore an oracle in the sense that it sees future demands and cannot be implemented in practice, and makes an assumption of stationarity. This metric follows the standard approach in online learning, which consists in comparing an algorithm to a constant strategy. A ratio below one therefore means that the algorithm considered has a better performance than $S_T^*$, and is equivalent to having a negative regret in online learning. In the appendix, we complete the results with two other metrics adapted to inventory problems, the lost sales and outdating percentage.

### 4.1 CYCLIC DEMANDS

We start with an experiment to illustrate the behavior of GAPSI against cyclic demands. We consider a single-product lost sales FIFO perishable inventory system without warehouse-capacity constraints ($K = 1, V_t = +\infty$). The demand of the product is given by the total demand of the M5 dataset described above, with a lifetime of two ($m = 2$) and no lead times ($L = 0$). We consider time-invariant unit costs.

To test the performance of GAPSI, we need to specify a choice of policy, that is, a choice of features. To see how the performances of GAPSI can be improved using features, we enhance GAPSI with 15 features as follows:

$$w_t = (D_T, D_T \mathbb{1}_{\{t\bmod 7=0\}}, \ldots, D_T \mathbb{1}_{\{t\bmod 7=6\}}, d_{t-7}, \ldots, d_{t-1}) \in \mathbb{R}_+^{15}, \tag{3}$$

where $D_T$ is an a priori upper bound on the demand. The first component of $w_t$ is time-invariant and plays the role of an intercept, the next 7 components can be interpreted as the one-hot encoding of the day of the week, and the last 7 components consist of past demands. We compare the performance of GAPSI with these features and GAPSI without any feature (in which case, only the intercept $D_T$ is kept in (3)). We also include in the comparison the best cyclic base-stock policy (each day of the week has its own base-stock level).

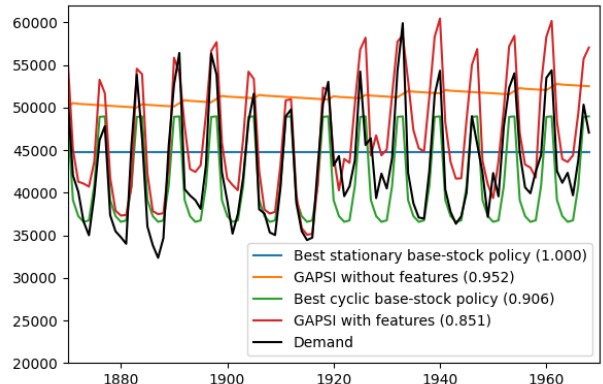

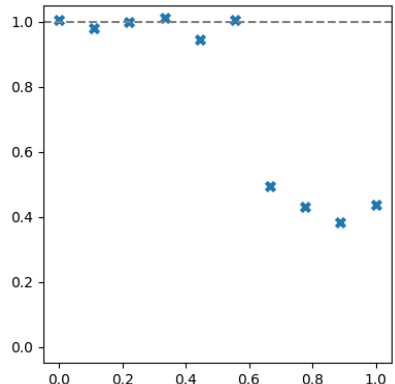

Figure 2: Demand (black curve) and base-stock levels (colored curves) of variants of GAPSI and baselines on a time window of 100 steps. In the legend, the ratio of losses is given between parenthesis for each algorithm.

Figure 3: Ratio of losses against level of variability (increasing from left to right) for 10 products selected from the M5 dataset (selected as quantiles of all products ranked by their normalized standard deviation)

The results are given in Figure 2. GAPSI without features outperforms the best stationary base-stock policy (it has a ratio of losses below 1). When considering the feature vector (3) we observe that these performances are further improved. The best cyclic base-stock policy positions itself between GAPSI without features and GAPSI with features. Figure 2 also shows the different behaviors: compared to the stationary base-stock policy (blue curve) GAPSI without features (orange curve) is able to adapt its level but have slow variations and cannot learn seasonal patterns unless features are provided (red curve).

## 4.2 COMPARISON STUDY

We now conduct an extensive comparison of GAPSI on several demand dynamics and against several competitors, in particular against MPC. MPC (Mattingley et al., 2011), also known as Receding Horizon Control, is a classical approach in operational research that is based on solving at each time period an optimization problem that aims at minimizing predicted future losses up to a receding planning horizon. Formally, at each time period $t$, using past information, we build a predictive model $\hat{x}_{\tau+1|t} = \hat{f}_{\tau|t}(\hat{x}_{\tau|t}, \hat{u}_{\tau|t})$ for $\tau = t, \ldots, t + H - 2$ that is initialized with $\hat{x}_{t|t} = x_t$ and minimize predicted losses $\sum_{\tau=t}^{t+H-1} \hat{\ell}_{\tau|t}(\hat{x}_{\tau|t}, \hat{u}_{\tau|t})$ under this predictive model. This provides a sequence of planned controls $\hat{u}_{t|t}, \ldots, \hat{u}_{t+H-1|t}$, from which we execute the control $u_t = \hat{u}_{t|t}$.

MPC therefore requires to have access to forecasted demands. To obtain a fair comparison, we thus take as features for GAPSI the same forecasts instead of the feature vector (3). We take as forecasts the demand of the previous week: $\hat{d}_t = d_{t-7}$. To obtain an estimate of the stability of the different algorithms with respect to noise in the forecasts, we perturb them with independent and identically distributed Gaussian noise, which yields $N = 10$ different forecasts.

We use the same inventory system as in the previous section and consider four different demand curves: three levels of the M5 dataset (Total, Category and Product), and the Total level of Califrais. Then, we compare the following algorithms: the MPC approach with a planning horizon of $H = 7$ days and planned demands equal to the forecasted demands $(\hat{d}_t, \ldots, \hat{d}_{t+6})$, the best stationary base-stock policy $S_T^*$, the non-stationary base-stock policies with levels $\hat{d}_t$, GAPSI without features, and GAPSI with features $w_t = (D_T, \hat{d}_t)$.

The results are given in Table 1. They show that both the base-stock policy with levels $\hat{d}_t$ and the MPC approach have similar performances. Both are significantly worse than the baseline $S_T^*$ and have the highest variance. On the other hand, GAPSI is able to outperform all the other algorithms while having a small variance. This experiment also shows that GAPSI can take advantage of

Table 1: Performances and robustness in terms of ratio of losses. Standard deviations are taken over 10 repetitions where we inject noise into the forecasts (GAPSI without forecasts therefore does not have standard deviations)

|  | M5 | | | Califrais |
|---|---|---|---|---|
|  | Total | Category | Product | Total |
| Base-stock levels $\hat{d}_t$ | $1.196 \pm 0.008$ | $1.219 \pm 0.009$ | $1.028 \pm 0.011$ | $1.579 \pm 0.022$ |
| MPC | $1.199 \pm 0.008$ | $1.219 \pm 0.009$ | $1.028 \pm 0.011$ | $1.579 \pm 0.021$ |
| GAPSI without forecasts | $0.952$ | $0.944$ | $0.735$ | $\mathbf{0.902}$ |
| GAPSI with forecasts | $\mathbf{0.910 \pm 0.002}$ | $\mathbf{0.904 \pm 0.002}$ | $\mathbf{0.704 \pm 0.005}$ | $0.912 \pm 0.003$ |

forecasts better than the other approaches tested which tend to "overfit" by ordering just enough to meet the forecasted demand. Moreover, MPC is slower to run (around 100 times longer), while base-stock level approaches have running times of the same order of magnitude as GAPSI. Complete running times are given in Appendix C

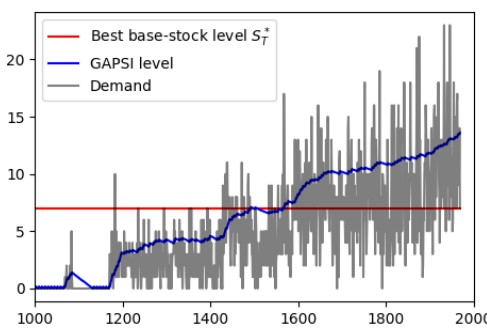

Figure 4: Base-stock levels of GAPSI and $S_T^*$ against the demand of product `HOUSEHOLD_1_022` whose normalized standard deviation is high (1.64) and corresponds to the ninth decile of all products. The last 969 time steps are shown.

Figure 5: Colored curves show the evolution of demands (solid curves) and GAPSI's base-stock levels (colored dashed curves) for each one of the 3 products (blue, yellow, green). Grey curves show the sum of GAPSI's base-stock levels (dashed curve) and the warehouse volume $V$ (solid line).

### 4.3 IMPACT OF THE VARIANCE

The larger the variability in the demands, the more difficult inventory optimisation becomes. Here we design an experiment to study the robustness of GAPSI to such variability. The parameters of this experiment are similar to the previous ones, the only differences being the demands, selected as follows. The 3049 products of the M5 dataset have been ranked in ascending order of variability. The variability is measured by their standard deviation over the horizon, normalized so that the magnitude of the demands does not affect this measure. We then select one product for each decile of the ranked products. The formula and examples of demands corresponding to different levels of variability can be found in Appendix C.

The results are given in Figures 3 and 4. In Figure 3, the ratio of losses is plotted against the level of variability, increasing from left to right. We notice that in the first six examples, which have the lowest normalized standard deviation, GAPSI incurs a cost that is close to the best stationary base-stock policy $S_T^*$, with a ratio of losses ranging from 0.944 to 1.013. Then, in the last 4 examples, which have a higher normalized variance, GAPSI drastically outperforms $S_T^*$ with a competitive ratio ranging from 0.384 to 0.493. This phenomenon is due to the nature of these last demands which feature many consecutive periods of zero demand, a situation in which GAPSI can adjust its base-stock level by temporarily reducing it, whereas the stationary policy $S_T^*$ cannot. Figure

4 is an example of such demand, and we can see how GAPSI adapts to a period of zero demand (between $t = 1000$ and $t = 1200$), and also how later, in a period of demand with a positive trend, it slowly increases its base-stock level. This illustrates how GAPSI is particularly well-adapted to non-stationary demands.

### 4.4 MULTIPLE PRODUCTS AND WAREHOUSE-CAPACITY CONSTRAINTS

We conclude with an experiment with multiple products and warehouse-capacity constraints. We consider a multi-product inventory system with $K = 3$ products with demands taken from the M5 dataset at the category level. The parameters of the experiment are similar to the previous experiment for each product, in particular we have $m_k = 3$ and $L_k = 0$, and GAPSI is run without features. The only difference is now the presence of a finite time-invariant warehouse volume $V_t = V < +\infty$, time-invariant unit volumes $v_{t,k} = 1$ and time-invariant overflow costs.

The results are given in Figure 5 where we see that GAPSI successfully satisfies the volume constraints and even saturates it which is the desired behavior since it minimizes lost sales. Indeed, the gray dashed curve is overall below the gray solid line, that is, $\sum_{k=1}^{K} v_{t,k} S_{t,k} = \sum_{k=1}^{K} S_{t,k} \lesssim V$. Notice that this is happening even though demand is overall increasing.

## 5 CONCLUSION

In this paper, we used techniques from online learning and insights from inventory control theory to address realistic inventory problems. We showed that the recent framework of Online Policy Selection (OPS) (Lin et al., 2024), at the crossroads of online learning and control, is well-adapted to model complex inventory problems involving, for instance, multiple products, perishability, order lead times and warehouse-capacity constraints. To address such problems, we designed GAPSI, a new online algorithm, and showed its efficiency through extensive numerical simulations.

However, the question of theoretical guarantees remains open. The standard assumptions of OPS are not satisfied for the inventory problems we consider. Indeed, policies, losses and transitions are not differentiable so that both classical chain rules and smoothness do not hold. Furthermore, the contraction property used in Lin et al. (2024) does not seem to hold. Proving regret bounds without these assumptions is very challenging.

The work of Bolte & Pauwels (2021) could be a promising way of handling the non-differentiability issue. They develop a flexible calculus theory for a new notion of generalized derivatives, allowing to consider chain rules for functions that are not differentiable in the classical sense.

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

# Supplementary Material for "Online policy selection for inventory problems"

## Table of Contents

## A  INVENTORY PROBLEMS

### A.1  COMPLETE EXPRESSION OF TRANSITION FUNCTIONS AND LOSSES

**Order lead times.** The transition function with order lead times writes

$$f_{t,i}(x_t, u_t) = \left[ z_{t,i+1} - \left[ d_t - \sum_{i'=1}^{i} z_{t,i'} \right]^+ \right]^+ \qquad \text{for } i = 1, \ldots, m-1,$$

$$f_{t,i}(x_t, u_t) = z_{t,i+1} \qquad \text{for } i = m, \ldots, m+L-1.$$

**Multi-product system with warehouse-capacity constraints.** Let us provide two examples of models handling warehouse-capacity constraints with multiple products. The first one is adapted to a simplified setting without lead-times and with non-perishable products. The second one is the one we consider. Assume there are $K \in \mathbb{N}$ product types indexed by $k \in [K]$.

- Shi et al. (2016) consider a lost sales system with instantaneous replenishment $(L = 0)$ and non-perishable products, that is, every product follows the following transition :

$$f_t(x_t, u_t) = \big( f_{t,k}(x_t, u_t) \big)_{k \in [K]} = \big( [x_{t,k} + u_{t,k} - d_{t,k}]^+ \big)_{k \in [K]}.$$

  However, instead of allowing the manager to choose arbitrary order quantities $u_t \in \mathbb{R}_+^K$, they are restricted to choose $u_t \in \mathbb{R}_+^K$ such that: $\sum_{k=1}^{K}(x_{t,k} + u_{t,k}) \leq V$, where $V > 0$

is the warehouse capacity. Notice that the constraint can be expressed as $z_t \in \mathbb{V}$, where $z_t = (x_{t,k}, u_{t,k})_{k \in [K]}$ and,

$$\mathbb{V} = \left\{ z_t = (x_{t,k}, u_{t,k})_{k \in [K]} \mid \sum_{k=1}^{K}(x_{t,k} + u_{t,k}) \le V \right\}.$$

- Consider the multi-product setting where each product $k \in [K]$ is perishable with lifetime $m_k \in \mathbb{N}$ and have a lead time $L_k \in \mathbb{N}_0$. The warehouse-capacity constraint is satisfied if and only if $z_t \in \mathbb{V}_t$, where

$$\mathbb{V}_t = \left\{ z_t \mid \sum_{k=1}^{K}\sum_{i=1}^{m_k} v_{t,k} z_{t,k,i} \le V_t \right\},$$

where for a time period $t$, $v_{t,k}$ is the unit volume of product $k$ and $V_t$ is the total volume of the warehouse. If the constraint is not satisfied, we remove products that just arrived in the ascending order (starting from product $k = 1, 2, \ldots, K$). Formally, this corresponds to defining $\tilde{z}_{t,k,m_k}$ as follows:

$$\left[ z_{t,k,m_k} - \frac{1}{v_{t,k}} \left[ \left[ \sum_{k'=1}^{K}\sum_{i'=1}^{m_{k'}} v_{t,k'} z_{t,k',i'} - V_t \right]^+ - \sum_{k'=1}^{k-1} v_{t,k'} z_{t,k',m_{k'}} \right]^+ \right]^+, \quad (4)$$

and $\tilde{z}_{t,k,i} = z_{t,k,i}$ for all $i \in [m_k + L_k] \setminus \{m_k\}$. The reader may find Equation (4) hard to interpret, so we provide in Appendix A.2 an alternative expression which is more easily interpretable.

Finally, to obtain the next state $x_{t+1}$ we simply need to evaluate, for every product $k$, the transition on the resulting vector $\tilde{z}_{t,k}$ rather than $z_{t,k}$. That is,

$$f_{t,k,i}(x_t, u_t) = \begin{cases} \left[ \tilde{z}_{t,k,i+1} - \left[ d_{t,k} - \sum_{i'=1}^{i} \tilde{z}_{t,k,i'} \right]^+ \right]^+ & \text{for } i = 1, \ldots, m_k - 1, \\ \tilde{z}_{t,k,i+1} & \text{for } i = m_k, \ldots, m_k + L_k - 1. \end{cases} \quad (5)$$

Notice that setting $V_t = +\infty$ removes the warehouse-capacity constraints. Finally, we assume that $m_k + L_k \ge 2$ for all products $k \in [K]$, that is, we never have both $m_k = 1$ and $L_k = 0$, allowing us to avoid dealing with the degenerate case where the dimension of a product's state is zero ($n_k = m_k + L_k - 1 = 0$).

**Loss.** The complete loss function writes as

$$\ell_t(x_t, u_t) = \sum_{k=1}^{K} \left( c_{t,k}^{\text{pena}} \cdot \left[ d_{t,k} - \sum_{i=1}^{m_k} \tilde{z}_{t,k,i} \right]^+ + c_{t,k}^{\text{hold}} \cdot \left[ \sum_{i=1}^{m_k} \tilde{z}_{t,k,i} - d_{t,k} \right]^+ + c_{t,k}^{\text{purc}} \cdot u_{t,k} \right.$$

$$\left. + c_{t,k}^{\text{outd}} \cdot [\tilde{z}_{t,k,1} - d_{t,k}]^+ + c_{t,k}^{\text{over}} \cdot (z_{t,k,m_k} - \tilde{z}_{t,k,m_k}) \right), \quad (6)$$

where $c_{t,k}^{\text{pena}}, c_{t,k}^{\text{hold}}, c_{t,k}^{\text{purc}}, c_{t,k}^{\text{outd}}, c_{t,k}^{\text{over}} \ge 0$ corresponds to unit costs for product $k$ at time period $t$, for each respective type of cost.

## A.2 EQUIVALENT EXPRESSIONS FOR DISCARDING

In this appendix we give an alternative expression for $\tilde{z}_t$, the state-control vector after discarding, defined in Equation (4) and provide its interpretation afterwards.

**Proposition A.1.** *Let $o_t$ denote the volume overflow defined as:*

$$o_t = \left[ \sum_{k'=1}^{K}\sum_{i'=1}^{m_{k'}} v_{t,k'} z_{t,k',i'} - V_t \right]^+. \quad (7)$$

*Also, define recursively the volume to remove for $k = 1, \ldots, K$,*

$$r_{t,k} = \min \left\{ v_{t,k} z_{t,k,m_k} , \left[ o_t - \sum_{k'=1}^{k-1} r_{t,k'} \right]^+ \right\}. \tag{8}$$

*Then, we have,*

$$\sum_{k'=1}^{k} r_{t,k'} = \min \left\{ \sum_{k'=1}^{k} v_{t,k'} z_{t,k',m_{k'}} , o_t \right\}. \tag{9}$$

*Furthermore, $\tilde{z}_{t,k,m_k}$ defined in (4), can be rewritten as:*

$$\tilde{z}_{t,k,m_k} = z_{t,k,m_k} - \frac{r_{t,k}}{v_{t,k}}. \tag{10}$$

*Proof.* We start by proving Equation (9) using an induction over $k = 1, \ldots, K$. First, for $k = 1$, we have:

$$\sum_{k'=1}^{1} r_{t,k'} = r_{t,1} \stackrel{(8)}{=} \min \left\{ v_{t,1} z_{t,1,m_1} , o_t \right\} = \min \left\{ \sum_{k'=1}^{1} v_{t,k'} z_{t,k',m_{k'}} , o_t \right\}.$$

Now, assume Equation (9) holds for some $k \in [K-1]$. Then,

$$\sum_{k'=1}^{k+1} r_{t,k'}$$

$$= \sum_{k'=1}^{k} r_{t,k'} + \min \left\{ v_{t,k+1} z_{t,k+1,m_{k+1}} , \left[ o_t - \sum_{k'=1}^{k} r_{t,k'} \right]^+ \right\}$$
$$\text{(using Equation (8))}$$

$$= \sum_{k'=1}^{k} r_{t,k'} + \min \left\{ v_{t,k+1} z_{t,k+1,m_{k+1}} , \underbrace{\left[ o_t - \min \left\{ \sum_{k'=1}^{k} v_{t,k'} z_{t,k',m_{k'}} , o_t \right\} \right]^+}_{\cdot \geq 0} \right\}$$
$$\text{(using Equation (9))}$$

$$= \sum_{k'=1}^{k} r_{t,k'} + \min \left\{ v_{t,k+1} z_{t,k+1,m_{k+1}} , o_t - \min \left\{ \sum_{k'=1}^{k} v_{t,k'} z_{t,k',m_{k'}} , o_t \right\} \right\}$$

$$= \min \left\{ \sum_{k'=1}^{k} r_{t,k'} + v_{t,k+1} z_{t,k+1,m_{k+1}} , \sum_{k'=1}^{k} r_{t,k'} + o_t - \min \left\{ \sum_{k'=1}^{k} v_{t,k'} z_{t,k',m_{k'}} , o_t \right\} \right\}$$

$$= \min \left\{ \min \left\{ \sum_{k'=1}^{k} v_{t,k'} z_{t,k',m_{k'}} , o_t \right\} + v_{t,k+1} z_{t,k+1,m_{k+1}} , o_t \right\}$$
$$\text{(using Equation (9))}$$

$$= \min \left\{ \sum_{k'=1}^{k+1} v_{t,k'} z_{t,k',m_{k'}} , o_t + v_{t,k+1} z_{t,k+1,m_{k+1}} , o_t \right\}$$

$$= \min \left\{ \sum_{k'=1}^{k+1} v_{t,k'} z_{t,k',m_{k'}} , o_t \right\},$$

where the last equality comes from the fact that $v_{t,k+1} z_{t,k+1,m_{k+1}} \geq 0$. This completes the proof of Equation (9).

Now we move to the proof of Equation (10).

$$z_{t,k,m_k} - \frac{r_{t,k}}{v_{t,k}} = z_{t,k,m_k} - \min\left\{ z_{t,k,m_k} , \frac{1}{v_{t,k}}\left[ o_t - \sum_{k'=1}^{k-1} r_{t,k'} \right]^+ \right\}$$

(using Equation (8))

$$= \left[ z_{t,k,m_k} - \frac{1}{v_{t,k}}\left[ o_t - \sum_{k'=1}^{k-1} r_{t,k'} \right]^+ \right]^+$$

$$= \left[ z_{t,k,m_k} - \frac{1}{v_{t,k}}\left[ o_t - \min\left\{ \sum_{k'=1}^{k-1} v_{t,k'} z_{t,k',m_{k'}} , o_t \right\} \right]^+ \right]^+$$

(using Equation (9))

$$= \left[ z_{t,k,m_k} - \frac{1}{v_{t,k}}\left[ o_t - \sum_{k'=1}^{k-1} v_{t,k'} z_{t,k',m_{k'}} \right]^+ \right]^+$$

$$= (4),$$

where we used in the second and fourth equality that $[a - b]^+ = a - \min\{a, b\}$, for any $a, b \in \mathbb{R}$. $\quad\square$

In light of Proposition A.1, it becomes easier to interpret the state-control vector after discarding $\tilde{z}_t$ defined by Equation (4). Indeed, at the reception of the products, one first checks the volume overflow $o_t$ defined by Equation (7). If $o_t = 0$ there is no overflow and we have $\tilde{z}_t = z_t$, otherwise we remove some products. In this case, we start discarding from the first product ($k = 1$). A volume $v_{t,1} z_{t,1,m_1}$ of product 1 just arrived to the warehouse and we remove a maximum of it to the extent of the volume overflow $o_t$, that is, we remove a volume $r_{t,1} = \min\{v_{t,1} z_{t,1,m_1}, o_t\}$ as defined in Equation (8). If $r_{t,1} = o_t$ then we do not discard products anymore, i.e. $r_{t,k} = 0$ for $k \in [K] \setminus \{1\}$. Otherwise, $r_{t,1} = v_{t,1} z_{t,1,m_1}$, and we move to the second product ($k = 2$) from which we want to remove a volume of $o_t - r_{t,1} \geq 0$ out of the volume $v_{t,2} z_{t,2,m_2}$ that just arrived, this defines $r_{t,2}$ according to Equation (8), and so on and so forth. Finally, from each product $k$ we removed a volume $r_{t,k}$ from the quantity $z_{t,k,m_k}$ and this completely defines $\tilde{z}_{t,k,m_k}$ through Equation (10).

## B  GAPSI

In this appendix we provide details on GAPSI. Section B.1 recalls how GAPS (Lin et al., 2024) and its gradient approximation procedure works. Then, Section B.2 motivates theoretically the use of custom derivative selections. Sections B.3 and B.4 state general definitions and properties related to one-sided derivatives and functions that are composed of positive parts. Finally, Sections B.5 and B.6 provide, in our new model, the formulas for the derivatives of the functions involved, in the general case and censored demand case respectively.

### B.1  THE GAPS ALGORITHM

**Main ideas.**   The GAPS algorithm (Lin et al., 2024) solves OPS problems using an online gradient descent approach with an approximated gradient. More precisely, define the surrogate functions $L_t : \Theta \to \mathbb{R}$ where $L_t(\theta)$ is the loss incurred at time period $t$ if we had followed the policy associated to $\theta$ for all periods, that is, if we applied the controls $u_s = \pi_s(x_s, \theta)$ for all $s = 1, \ldots, t$. An idealized gradient descent would be with respect to these losses $(L_t)_{t \geq 1}$, taking the form:

$$\theta_{t+1} = \mathrm{Proj}_\Theta(\theta_t - \eta_t \nabla L_t(\theta_t))$$

where $\Theta$ is the set of parameters, $\mathrm{Proj}_\Theta$ denotes the Euclidean projection operator on $\Theta$ and $\eta_t$ are learning rates.

However, the complexity of computing the gradients of $L_t$ exactly grows proportionally to $t$, becoming intractable when the horizon is large. In GAPS, two approximations are made to compute an approximated gradient:

1. Instead of computing the gradient $\nabla L_t(\theta_t)$ along the ideal trajectory $\theta_t, \ldots, \theta_t$, it is computed along the current trajectory $\theta_1, \ldots, \theta_t$.

2. The historical dependence is truncated to the $B$ most recent time steps.

**Implementation.** In practice, the approximated gradient can be computed in an online fashion using chain rules as detailed in Lin et al. (2024, Appendix B, Algorithm 2).

Since $L_t(\theta_t) = \ell_t(x_t(\theta_t), u_t(\theta_t))$ where $x_t(\theta)$ and $u_t(\theta) = \pi_t(x_t(\theta), \theta)$ denote the state and control if we had applied the parameter $\theta$ from period 1 to $t$, then,

$$\frac{\partial L_t(\theta_t)}{\partial \theta_t} = \left( \frac{\partial \ell_t(x_t(\theta_t), u_t(\theta_t))}{\partial x_t} + \frac{\partial \ell_t(x_t(\theta_t), u_t(\theta_t))}{\partial u_t} \cdot \frac{\partial \pi_t(x_t(\theta_t), \theta_t)}{\partial x_t} \right) \cdot \frac{\partial x_t(\theta_t)}{\partial \theta_t}$$
$$+ \frac{\partial \ell_t(x_t(\theta_t), u_t(\theta_t))}{\partial u_t} \cdot \frac{\partial \pi_t(x_t(\theta_t), \theta_t)}{\partial \theta_t}.$$

Applying the first approximation, that is, replacing $x_t(\theta_t)$ and $u_t(\theta_t)$ by $x_t$ and $u_t$ respectively, we obtain a first expression for the approximated gradient:

$$\left( \frac{\partial \ell_t(x_t, u_t)}{\partial x_t} + \frac{\partial \ell_t(x_t, u_t)}{\partial u_t} \cdot \frac{\partial \pi_t(x_t, \theta_t)}{\partial x_t} \right) \cdot \frac{\partial x_t(\theta_t)}{\partial \theta_t} + \frac{\partial \ell_t(x_t, u_t)}{\partial u_t} \cdot \frac{\partial \pi_t(x_t, \theta_t)}{\partial \theta_t}.$$

Now, let us deal with $\partial x_t(\theta_t)/\partial \theta_t$. Since $x_t(\theta_t) = f_{t-1}(x_{t-1}(\theta_t), u_{t-1}(\theta_t))$, we have that:

$$\frac{\partial x_t(\theta_t)}{\partial \theta_t} = \left( \frac{\partial f_{t-1}(x_{t-1}(\theta_t), u_{t-1}(\theta_t))}{\partial x_{t-1}} + \frac{\partial f_{t-1}(x_{t-1}(\theta_t), u_{t-1}(\theta_t))}{\partial u_{t-1}} \cdot \frac{\partial \pi_{t-1}(x_{t-1}(\theta_t), \theta_t)}{\partial x_{t-1}} \right) \cdot \frac{\partial x_{t-1}(\theta_t)}{\partial \theta_{t-1}}$$
$$+ \frac{\partial f_{t-1}(x_{t-1}(\theta_t), u_{t-1}(\theta_t))}{\partial u_{t-1}} \cdot \frac{\partial \pi_{t-1}(x_{t-1}(\theta_t), \theta_t)}{\partial \theta_{t-1}}.$$

Applying the first approximation, that is, replacing the points $x_{t-1}(\theta_t)$, $u_{t-1}(\theta_t)$ and $\theta_t$ by $x_{t-1}$, $u_{t-1}$ and $\theta_{t-1}$ respectively, we obtain the following approximating expression for $\partial x_t(\theta_t)/\partial \theta_t$,

$$\left( \frac{\partial f_{t-1}(x_{t-1}, u_{t-1})}{\partial x_{t-1}} + \frac{\partial f_{t-1}(x_{t-1}, u_{t-1})}{\partial u_{t-1}} \cdot \frac{\partial \pi_{t-1}(x_{t-1}, \theta_{t-1})}{\partial x_{t-1}} \right) \cdot \frac{\partial x_{t-1}(\theta_{t-1})}{\partial \theta_{t-1}}$$
$$+ \frac{\partial f_{t-1}(x_{t-1}, u_{t-1})}{\partial u_{t-1}} \cdot \frac{\partial \pi_{t-1}(x_{t-1}, \theta_{t-1})}{\partial \theta_{t-1}}.$$

Together with the second approximation, this expression leads to a recursive way of approximating $\partial x_t(\theta_t)/\partial \theta_t$ and thus $\nabla L_t(\theta_t)$ which is detailed below in Algorithm 2 with our notations (see also Lin et al. (2024, Appendix B, Algorithm 2)).

---

**Algorithm 2:** Gradient approximation procedure

1   $\frac{\partial x_t}{\partial \theta_{t-1}} \leftarrow \frac{\partial f_{t-1}}{\partial u_{t-1}} \cdot \frac{\partial \pi_{t-1}}{\partial \theta_{t-1}}$;

2   **for** $b = 2, \ldots, B-1$ **do**

3     $\left\lfloor \; \frac{\partial x_t}{\partial \theta_{t-b}} \leftarrow \left( \frac{\partial f_{t-1}}{\partial x_{t-1}} + \frac{\partial f_{t-1}}{\partial u_{t-1}} \cdot \frac{\partial \pi_{t-1}}{\partial x_{t-1}} \right) \cdot \frac{\partial x_{t-1}}{\partial \theta_{t-b}} \right.$;

4   **return** $\frac{\partial \ell_t}{\partial u_t} \cdot \frac{\partial \pi_t}{\partial \theta_t} + \left( \frac{\partial \ell_t}{\partial x_t} + \frac{\partial \ell_t}{\partial u_t} \cdot \frac{\partial \pi_t}{\partial x_t} \right) \cdot \sum_{b=1}^{B-1} \frac{\partial x_t}{\partial \theta_{t-b}}$;

---

**Theoretical guarantees.** Under suitable assumptions Lin et al. (2024) are able to prove theoretical guarantees for GAPS in the form of regret upper bounds.

Assuming, amongst other things, Lipschitz continuity, smoothness, a contractive perturbation property, the convexity of $L_t$, a large enough horizon $T$ and buffer length $B$ and appropriately tuned learning rates, Corollary 3.4 of Lin et al. (2024) gives in particular for any $\theta \in \Theta$,

$$\sum_{t=1}^{T} \ell_t(x_t, u_t) - L_t(\theta) = O(\sqrt{T}),$$

where $O(\cdot)$ hides problem-dependent constants.

## B.2 ON THE SELECTION OF DERIVATIVES

In this appendix, we illustrate on a simple theoretical example a problematic behavior of GAPSI when the derivatives or not carefully selected.

**Proposition B.1.** *Consider any single-product inventory problem without dynamics and assume GAPSI follows a base-stock policy. Assume further that it is implemented using auto-differentiation with $ReLU'(0) = 0$ or using left derivatives for the policy. Then, if the base-stock level reaches zero it remains so for all the subsequent periods.*

*Proof.* Consider a single-product inventory problem with scalar states, controls and parameters where there are no dynamics ($f_t \equiv 0$) and arbitrary losses $\ell_t$. Assume GAPSI follows the base-stock policy: $\pi_t(x_t, \theta_t) = [\theta_t - x_t]^+$ over $\Theta = [0, 1]$ and that we set $\frac{\partial \pi_t}{\partial \theta_t}(x_t, \theta_t) \leftarrow \mathbb{1}_{\{\theta_t > x_t\}}$ which holds if auto-differentiation is used with $ReLU'(0) = 0$ or if left derivatives are used for the policy.

Since there are no dynamics, following line 4 of Algorithm 2 (or line 10 of Algorithm 2 in Appendix B of (Lin et al., 2024)), we have:

$$g_t = \frac{\partial \ell_t}{\partial u_t}(x_t, u_t) \cdot \frac{\partial \pi_t}{\partial \theta_t}(x_t, \theta_t) = \frac{\partial \ell_t}{\partial u_t}(x_t, u_t) \cdot \mathbb{1}_{\{\theta_t > x_t\}}$$

Therefore for any $t \in \mathbb{N}$, $\theta_t = 0$ implies $g_t = 0$, which in turn implies $\theta_{t+1} = \text{Proj}_\Theta(\theta_t - H_t g_t) = 0$ and so on and so forth. $\square$

The undesirable stationary behavior described in Proposition B.1 can also be observed in practice (see Figure 1). One may wonder whether this is happening because of a wrong application of the chain rule (which is applied here on functions that are not differentiable everywhere). In fact, this is not the case, since according to the base-stock policy $\pi_t(x_t, \theta_t) = 0$ for all $\theta_t \leq x_t$, this behavior still happens even if we computed the exact left derivative of $\tilde{\ell}_t(\theta) = \ell_t(x_t, \pi_t(x_t, \theta))$ and performed an (exact) online subgradient descent with respect to these derivatives. Notice that even if $\ell_t(x_t, \cdot)$ is convex, $\tilde{\ell}_t$ is not necessarily convex which explains the difficulties encountered by a subgradient descent. Due to this undesirable behavior we advocate for the use of right derivatives instead of left derivatives for the policies.

## B.3 DEFINITIONS AND PROPERTIES OF ONE-SIDED DERIVATIVES

First, we start by some definitions regarding one-sided and standard derivatives.

**Definition B.2.** Let $f : \mathbb{R} \to \mathbb{R}$ and $x \in \mathbb{R}$. We denote by

$$\partial^+ f(x) = \lim_{h \to 0^+} \frac{f(x+h) - f(x)}{h}$$

whenever it exists and say that $f$ is *right-differentiable* at $x$ in such a case. Similarly, we denote by

$$\partial^- f(x) = \lim_{h \to 0^-} \frac{f(x+h) - f(x)}{h}$$

whenever it exists and say that $f$ is *left-differentiable* at $x$ in such a case. Finally, we denote by

$$\partial f(x) = \lim_{h \to 0} \frac{f(x+h) - f(x)}{h}$$

whenever it exists and say that $f$ is *differentiable* at $x$ in such a case.

One-sided derivatives and classical derivatives are related by the following proposition.

**Proposition B.3.** *Let $f : \mathbb{R} \to \mathbb{R}$ and $x \in \mathbb{R}$. $f$ is differentiable at $x$ if and only if $f$ is right-differentiable at $x$ and left-differentiable at $x$ with $\partial^+ f(x) = \partial^- f(x)$. In such a case, we have $\partial f(x) = \partial^+ f(x) = \partial^- f(x)$.*

The following proposition is concerned by the one-sided derivatives of a composition of functions involving the positive part.

**Proposition B.4.** *Let $f : \mathbb{R} \to \mathbb{R}$ be a function that is continuous at $0$ and define for all $h \in \mathbb{R}$, $g(h) = [f(h)]^+$.*

- *If $f(0) < 0$, then, $g$ is differentiable at $0$ and $\partial g(0) = 0$.*

- *If $f(0) = 0$ and $f$ is right-differentiable at $0$, then, $g$ is right-differentiable at $0$ and $\partial^+ g(0) = [\partial^+ f(0)]^+$.*

- *If $f(0) = 0$ and $f$ is left-differentiable at $0$, then, $g$ is left-differentiable at $0$ and $\partial^- g(0) = -[-\partial^- f(0)]^+$.*

- *If $f(0) > 0$ and $f$ is right-differentiable at $0$, then, $g$ is right-differentiable at $0$ and $\partial^+ g(0) = \partial^+ f(0)$.*

- *If $f(0) > 0$ and $f$ is left-differentiable at $0$, then, $g$ is left-differentiable at $0$ and $\partial^- g(0) = \partial^- f(0)$.*

*Proof.*

- First, let us assume $f(0) < 0$, thus $g(0) = 0$. Since $f$ is continuous at $0$, there exists $\varepsilon > 0$, such that $f(h) < 0$ for all $h \in (-\varepsilon, \varepsilon)$. Therefore, $g(h) = [f(h)]^+ = 0$ for all $h \in (-\varepsilon, \varepsilon)$. In particular, $\partial g(0) = \lim_{h \to 0} (g(h) - g(0))/h$ exists and is equal to $0$.

- Now, assume $f(0) = 0$, thus $g(0) = 0$. Then for all $h > 0$, we have $g(h)/h = [f(h)]^+/h = [f(h)/h]^+$ and since the positive part is a continuous function, we have $\lim_{h \to 0^+} g(h)/h = [\lim_{h \to 0^+} f(h)/h]^+$ as soon as $\lim_{h \to 0^+} f(h)/h$ exists. Similarly, for all $h < 0$ we have $g(h)/h = [f(h)]^+/h = -[-f(h)/h]^+$ and since the positive part and $h \mapsto -h$ are continuous functions, we have $\lim_{h \to 0^-} g(h)/h = -[-\lim_{h \to 0^-} f(h)/h]^+$ as soon as $\lim_{h \to 0^-} f(h)/h$ exists.

- Finally, assume $f(0) > 0$. Since $f$ is continuous at $0$, there exists $\varepsilon > 0$, such that $f(h) > 0$ for all $h \in (-\varepsilon, \varepsilon)$. Therefore, $g(h) = [f(h)]^+ = f(h)$ for all $h \in (-\varepsilon, \varepsilon)$. In particular, if $\lim_{h \to 0^+} (f(h) - f(0))/h$ exists then $\lim_{h \to 0^+} (g(h) - g(0))/h$ also exists and they are both equal. The same claim holds for $\lim_{h \to 0^-} (f(h) - f(0))/h$ and $\lim_{h \to 0^-} (g(h) - g(0))/h$.

$\square$

The following corollary gives a convenient way to compute one-sided derivatives of compositions involving the positive part.

**Corollary B.5.** *Let $x \in \mathbb{R}$ and $f : \mathbb{R} \to \mathbb{R}$ a function that is right-differentiable at $x$ and left-differentiable at $x$. Then, the function $g$ defined by $g(h) = [f(h)]^+$ for all $h \in \mathbb{R}$, is continuous at $x$, right-differentiable at $x$ and left-differentiable at $x$, furthermore,*

$$\partial^+ g(x) = \left(\partial^+ f(x)\right)\left(\mathbb{1}_{\{f(x)>0\}} + \mathbb{1}_{\{\partial^+ f(x)>0\}}\mathbb{1}_{\{f(x)=0\}}\right),$$

$$\partial^- g(x) = \left(\partial^- f(x)\right)\left(\mathbb{1}_{\{f(x)>0\}} + \mathbb{1}_{\{\partial^- f(x)<0\}}\mathbb{1}_{\{f(x)=0\}}\right).$$

*Proof.* One can easily see that a function that is right-differentiable and left-differentiable at some point is necessarily continuous at this point. The rest of this corollary follows immediately from Proposition B.4. $\square$

The following example considers the case of an affine function composed by the positive part. It will be used repeatedly in the following.

**Example B.6.** Let $a, b \in \mathbb{R}$ and define the function $g$ as $g(x) = [ax + b]^+$ for all $x \in \mathbb{R}$. Then, $g$ is left-differentiable and right-differentiable for all $x \in \mathbb{R}$ and,

$$\partial^+ g(x) = a(\mathbb{1}_{\{ax+b>0\}} + \mathbb{1}_{\{a>0\}}\mathbb{1}_{\{ax+b=0\}}),$$

$$\partial^- g(x) = a(\mathbb{1}_{\{ax+b>0\}} + \mathbb{1}_{\{a<0\}}\mathbb{1}_{\{ax+b=0\}}).$$

*Proof.* This follows from Corollary B.5 when applied to $f(x) = ax + b$. $\square$

## B.5 ONE-SIDED DERIVATIVES FOR OUR MODEL

Before providing the one-sided derivatives of each component of the model: the policy, the losses and the transitions, we recall the notations used.

- Time is indexed by $t \in \mathbb{N}$ and product types are indexed by $k \in [K]$.

- The states are denoted by $x_t = (x_{t,k,i})_{k \in [K], i \in [n_k]} \in \mathbb{R}^n$ with $n = \sum_{k=1}^K n_k$ and $n_k = m_k + L_k - 1$ where $m_k$ and $L_k$ are respectively the lifetime and lead time of product $k$. The controls are denoted by $u_t = (u_{t,k})_{k \in [K]} \in \mathbb{R}^K$. State-controls are denoted $z_t = (x_{t,k,1} \ldots, x_{t,k,n_k}, u_{t,k})_{k \in [K]} \in \mathbb{R}^{n+K}$.

- The parameters are denoted by $\theta_t = (\theta_{t,k,i})_{k \in [K], i \in [p_k]} \in \mathbb{R}^P$ and the features are denoted by $w_t = (w_{t,k,i})_{k \in [K], i \in [p_k]} \in \mathbb{R}^P$ where $P = \sum_{k=1}^K p_k$.

- All these quantities are related in our model through the transitions $f_t$ defined in Equation (5) and the policy $\pi_t$ defined in Equation (3.1).

- Finally, the loss functions $\ell_t$ are defined in Equation (6).

### B.5.1 POLICY

The following proposition give the formulas for the one-sided partial derivatives of the feature-enhanced base-stock policy.

**Proposition B.7.** *Consider the feature-enhanced base-stock policy defined in Equation* (3.1) *and recalled here:*

$$\pi_{t,k}(x_t, \theta_t) = \left[ w_{t,k}^\top \theta_{t,k} - \sum_{i=1}^{n_k} x_{t,k,i} \right]^+.$$

*The right partial derivatives of the policy are given by:*

$$\frac{\partial^+ \pi_{t,k}(x_t, \theta_t)}{\partial x_{t,k',i'}} = -\mathbb{1}_{\{k=k'\}}\mathbb{1}_{\{w_{t,k}^\top \theta_{t,k} > \sum_{i=1}^{n_k} x_{t,k,i}\}},$$

$$\frac{\partial^+ \pi_{t,k}(x_t, \theta_t)}{\partial \theta_{t,k',i'}} = w_{t,k,i'}\mathbb{1}_{\{k=k'\}}\left( \mathbb{1}_{\{w_{t,k}^\top \theta_{t,k} > \sum_{i=1}^{n_k} x_{t,k,i}\}} + \mathbb{1}_{\{w_{t,k,i'}>0\}}\mathbb{1}_{\{w_{t,k}^\top \theta_{t,k} = \sum_{i=1}^{n_k} x_{t,k,i}\}} \right).$$

*The left partial derivatives of the policy are given by:*

$$\frac{\partial^- \pi_{t,k}(x_t, \theta_t)}{\partial x_{t,k',i'}} = -\mathbb{1}_{\{k=k'\}}\mathbb{1}_{\{w_{t,k}^\top \theta_{t,k} \geq \sum_{i=1}^{n_k} x_{t,k,i}\}},$$

$$\frac{\partial^- \pi_{t,k}(x_t, \theta_t)}{\partial \theta_{t,k',i'}} = w_{t,k,i'}\mathbb{1}_{\{k=k'\}}\left( \mathbb{1}_{\{w_{t,k}^\top \theta_{t,k} > \sum_{i=1}^{n_k} x_{t,k,i}\}} + \mathbb{1}_{\{w_{t,k,i'}<0\}}\mathbb{1}_{\{w_{t,k}^\top \theta_{t,k} = \sum_{i=1}^{n_k} x_{t,k,i}\}} \right).$$

*Proof.* For each $t \in \mathbb{N}$, $k, k' \in [K]$, $i \in [n_k]$, $i' \in [n_{k'}]$, consider the functions:

$$x_{t,k',i'} \mapsto w_{t,k}^\top \theta_{t,k} - \sum_{i=1}^{n_k} x_{t,k,i} \text{ and } \theta_{t,k',i'} \mapsto w_{t,k}^\top \theta_{t,k} - \sum_{i=1}^{n_k} x_{t,k,i}.$$

These are univariate real-valued *affine* functions with respective derivatives:

$$-\mathbb{1}_{\{k=k'\}} \text{ and } \mathbb{1}_{\{k=k'\}} w_{t,k,i'}.$$

Applying Example B.6 for these two functions leads to the desired result. $\square$

### B.5.2 STATE-CONTROL AFTER DISCARDING

Before moving to the derivations of one-sided partial derivatives of the losses and transitions which are more involved due to multiple compositions, we start by computing the derivatives of an auxiliary function: the state-control couple after discarding $\tilde{z}_t$ defined in Equation (4), with respect to the state-control couple $z_t$.

**Proposition B.8.** *Consider the vector the state-control couple after discarding $\tilde{z}_t \in \mathbb{R}^{n+K}$ which definition is provided in Equation (4) and recalled here.*

*Given the state-control couple $z_t \in \mathbb{R}^{n+K}$, we have $\tilde{z}_{t,k,m_k}$ equal to,*

$$
\left[ z_{t,k,m_k} - \frac{1}{v_{t,k}} \left[ \left[ \sum_{k''=1}^{K} \sum_{i''=1}^{m_{k''}} v_{t,k''} z_{t,k'',i''} - V_t \right]^+ - \sum_{k''=1}^{k-1} v_{t,k''} z_{t,k'',m_{k''}} \right]^+ \right]^+ ,
$$

*and $\tilde{z}_{t,k,i} = z_{t,k,i}$ for all $i \in [n_k+1] \setminus \{m_k\}$.*

*If $i \in [n_k+1] \setminus \{m_k\}$, we have:*

$$
\frac{\partial \tilde{z}_{t,k,i}}{\partial z_{t,k',i'}} = \mathbb{1}_{\{k=k'\}} \mathbb{1}_{\{i=i'\}}.
$$

*Now assume $i = m_k$, then, the one-sided partial derivatives are given by:*

$$
\frac{\partial^+ \tilde{z}_{t,k,i}}{\partial z_{t,k',i'}} = \partial^+ \alpha \cdot (\mathbb{1}_{\{\alpha>0\}} + \mathbb{1}_{\{\partial^+\alpha>0\}} \mathbb{1}_{\{\alpha=0\}}),
$$

$$
\frac{\partial^- \tilde{z}_{t,k,i}}{\partial z_{t,k',i'}} = \partial^- \alpha \cdot (\mathbb{1}_{\{\alpha>0\}} + \mathbb{1}_{\{\partial^-\alpha>0\}} \mathbb{1}_{\{\alpha=0\}}),
$$

*where:*

$$
\alpha = z_{t,k,i} - \frac{1}{v_{t,k}} [\beta]^+ ,
$$

$$
\beta = \left[ \sum_{k''=1}^{K} \sum_{i''=1}^{m_{k''}} v_{t,k''} z_{t,k'',i''} - V_t \right]^+ - \sum_{k''=1}^{k-1} v_{t,k''} z_{t,k'',m_{k''}},
$$

$$
\partial^+ \alpha = \mathbb{1}_{\{k=k'\}} \mathbb{1}_{\{i=i'\}} - \frac{1}{v_{t,k}} \partial^+ \beta \cdot \left( \mathbb{1}_{\{\beta>0\}} + \mathbb{1}_{\{\partial^+\beta>0\}} \mathbb{1}_{\{\beta=0\}} \right),
$$

$$
\partial^+ \beta = v_{t,k'} \left( \mathbb{1}_{\{i'\in[m_{k'}]\}} \mathbb{1}_{\{\sum_{k''=1}^{K} \sum_{i''=1}^{m_{k''}} v_{t,k''} z_{t,k'',i''} \geq V_t\}} - \mathbb{1}_{\{k'\in[k-1]\}} \mathbb{1}_{\{i'=m_{k'}\}} \right),
$$

$$
\partial^- \alpha = \mathbb{1}_{\{k=k'\}} \mathbb{1}_{\{i=i'\}} - \frac{1}{v_{t,k}} \partial^- \beta \cdot \left( \mathbb{1}_{\{\beta>0\}} + \mathbb{1}_{\{\partial^-\beta<0\}} \mathbb{1}_{\{\beta=0\}} \right),
$$

$$
\partial^- \beta = v_{t,k'} \left( \mathbb{1}_{\{i'\in[m_{k'}]\}} \mathbb{1}_{\{\sum_{k''=1}^{K} \sum_{i''=1}^{m_{k''}} v_{t,k''} z_{t,k'',i''} > V_t\}} - \mathbb{1}_{\{k'\in[k-1]\}} \mathbb{1}_{\{i'=m_{k'}\}} \right).
$$

*Proof.* The case $i \in [n_k+1] \setminus \{m_k\}$ is trivial. On the other hand, the formulas for the case $i = m_k$ are obtained by applying repeatedly Corollary B.5. First, consider the univariate real-valued affine function:

$$
z_{t,k',i'} \mapsto \sum_{k''=1}^{K} \sum_{i''=1}^{m_{k''}} v_{t,k''} z_{t,k'',i''} - V_t.
$$

and apply Corollary B.5 (or Example B.6). Then, we derive easily $\partial^+\beta$ and $\partial^-\beta$. Applying Corollary B.5 to $\beta$, leads us easily to $\partial^+\alpha$ and $\partial^-\alpha$. A final application of Corollary B.5 to $\alpha$ allows us to conclude. $\square$

### B.5.3 LOSSES

In the following we give, in the general case, the one-sided partial derivatives of the loss functions used of our model, defined in Equation (6).

**Proposition B.9.** *Consider the loss functions of our model defined in Equation* (6) . *The one-sided partial derivatives of the losses are given by the following, for each* $\square \in \{-,+\}$,

$$
\begin{aligned}
\frac{\partial^{\square} \ell_t(z_t)}{\partial z_{t,k',i'}} = \sum_{k=1}^{K} \Bigg( & c_{t,k}^{\text{purc}} \mathbb{1}_{\{k=k'\}} \mathbb{1}_{\{i'=m_{k'}+L_{k'}\}} \\
& + c_{t,k}^{\text{hold}} \partial^{\square} \gamma^{\text{hold}} \cdot \left( \mathbb{1}_{\{\sum_{i=1}^{m_k} \tilde{z}_{t,k,i} > d_{t,k}\}} + \mathbb{1}_{\{\square \cdot \partial^{\square} \gamma^{\text{hold}} > 0\}} \mathbb{1}_{\{\sum_{i=1}^{m_k} \tilde{z}_{t,k,i} = d_{t,k}\}} \right) \\
& - c_{t,k}^{\text{pena}} \partial^{\square} \gamma^{\text{hold}} \cdot \left( \mathbb{1}_{\{\sum_{i=1}^{m_k} \tilde{z}_{t,k,i} < d_{t,k}\}} + \mathbb{1}_{\{\square \cdot \partial^{\square} \gamma^{\text{hold}} < 0\}} \mathbb{1}_{\{\sum_{i=1}^{m_k} \tilde{z}_{t,k,i} = d_{t,k}\}} \right) \\
& + c_{t,k}^{\text{outd}} \partial^{\square} \gamma^{\text{outd}} \cdot \left( \mathbb{1}_{\{\tilde{z}_{t,k,1} > d_{t,k}\}} + \mathbb{1}_{\{\square \cdot \partial^{\square} \gamma^{\text{outd}} > 0\}} \mathbb{1}_{\{\tilde{z}_{t,k,1} = d_{t,k}\}} \right) \\
& + c_{t,k}^{\text{over}} \cdot \left( \mathbb{1}_{\{k=k'\}} \mathbb{1}_{\{i'=m_{k'}\}} - \frac{\partial^{\square} \tilde{z}_{t,k,m_k}}{\partial z_{t,k',i'}} \right) \Bigg),
\end{aligned}
$$

*where:*

$$
\partial^{\square} \gamma^{\text{hold}} = \frac{\partial^{\square} \tilde{z}_{t,k,m_k}}{\partial z_{t,k',i'}} + \mathbb{1}_{\{k'=k\}} \mathbb{1}_{\{i' \in [m_{k'}-1]\}},
$$

$$
\partial^{\square} \gamma^{\text{outd}} = \mathbb{1}_{\{m_k=1\}} \frac{\partial^{\square} \tilde{z}_{t,k,m_k}}{\partial z_{t,k',i'}} + \mathbb{1}_{\{m_k \neq 1\}} \mathbb{1}_{\{k'=k\}} \mathbb{1}_{\{i'=m_{k'}\}},
$$

*and* $\partial^{\square} \tilde{z}_{t,k,m_k} / \partial z_{t,k',i'}$ *is given by Proposition B.8.*

*Proof.* This follows from Proposition B.8 and Corollary B.5. $\qquad\square$

### B.5.4 TRANSITIONS

In the following we give, in the general case, the one-sided partial derivatives of the transition functions of our model, defined in Equation (5).

**Proposition B.10.** *Consider the transition functions of our model defined in Equation* (5) *and recalled here:*

$$
f_{t,k,i}(z_t) = \begin{cases} \left[ \tilde{z}_{t,k,i+1} - \left[ d_{t,k} - \sum_{i'=1}^{i} \tilde{z}_{t,k,i'} \right]^+ \right]^+ & \text{for } i = 1, \ldots, m_k - 1, \\ \tilde{z}_{t,k,i+1} & \text{for } i = m_k, \ldots, m_k + L_k - 1. \end{cases}
$$

*where* $\tilde{z}_t$ *is the state-control couple after discarding defined in Equation* (4).

*If* $i \in [n_k] \setminus \{m_k - 1\}$, *then, the one-sided derivatives of the transition functions are given by:*

$$
\frac{\partial^+ f_{t,k,i}(z_t)}{z_{t,k',i'}} = \mathbb{1}_{\{k=k'\}} \Big( \mathbb{1}_{\{i \in [n_k] \setminus [m_k-1]\}} \mathbb{1}_{\{i'=i+1\}} + \mathbb{1}_{\{i \in [m_k-1]\}} \big(
$$

$$
\mathbb{1}_{\{i'=i+1\}} \mathbb{1}_{\{z_{t,k,i+1} \geq [d_{t,k} - \sum_{i''=1}^{i} z_{t,k,i''}]^+\}} + \mathbb{1}_{\{i' \in [i]\}} \mathbb{1}_{\{z_{t,k,i+1} \geq d_{t,k} - \sum_{i''=1}^{i} z_{t,k,i''} > 0\}} \big) \Big),
$$

$$
\frac{\partial^- f_{t,k,i}(z_t)}{z_{t,k',i'}} = \mathbb{1}_{\{k=k'\}} \Big( \mathbb{1}_{\{i \in [n_k] \setminus [m_k-1]\}} \mathbb{1}_{\{i'=i+1\}} + \mathbb{1}_{\{i \in [m_k-1]\}} \big(
$$

$$
\mathbb{1}_{\{i'=i+1\}} \mathbb{1}_{\{z_{t,k,i+1} > [d_{t,k} - \sum_{i''=1}^{i} z_{t,k,i''}]^+\}} + \mathbb{1}_{\{i' \in [i]\}} \mathbb{1}_{\{z_{t,k,i+1} > d_{t,k} - \sum_{i''=1}^{i} z_{t,k,i''} \geq 0\}} \big) \Big).
$$

*If* $i = m_k - 1$, *then, the one-sided derivatives of the transition functions are given by:*

$$
\frac{\partial^+ f_{t,k,i}(z_t)}{z_{t,k',i'}} = \partial^+ \delta \cdot \left( \mathbb{1}_{\{\delta > 0\}} + \mathbb{1}_{\{\partial^+ \delta > 0\}} \mathbb{1}_{\{\delta = 0\}} \right)
$$

$$
\frac{\partial^- f_{t,k,i}(z_t)}{z_{t,k',i'}} = \partial^- \delta \cdot \left( \mathbb{1}_{\{\delta > 0\}} + \mathbb{1}_{\{\partial^- \delta < 0\}} \mathbb{1}_{\{\delta = 0\}} \right),
$$

*where,*

$$\delta = \tilde{z}_{t,k,m_k} - \left[ d_{t,k} - \sum_{i''=1}^{m_k-1} z_{t,k,i''} \right]^+,$$

$$\partial^+ \delta = \frac{\partial^+ \tilde{z}_{t,k,m_k}}{\partial z_{t,k',i'}} + \mathbb{1}_{\{k'=k\}} \mathbb{1}_{\{i' \in [m_{k'}-1]\}} \mathbb{1}_{\{d_{t,k} > \sum_{i''=1}^{m_k-1} z_{t,k,i''}\}},$$

$$\partial^- \delta = \frac{\partial^- \tilde{z}_{t,k,m_k}}{\partial z_{t,k',i'}} + \mathbb{1}_{\{k'=k\}} \mathbb{1}_{\{i' \in [m_{k'}-1]\}} \mathbb{1}_{\{d_{t,k} \geq \sum_{i''=1}^{m_k-1} z_{t,k,i''}\}},$$

*and $\partial^+ \tilde{z}_{t,k,m_k}/\partial z_{t,k',i'}$ and $\partial^- \tilde{z}_{t,k,m_k}/\partial z_{t,k',i'}$ are given by Proposition B.8.*

*Proof.* To prove this proposition, we start by recalling that $\tilde{z}_{t,k,i} = z_{t,k,i}$ for all indexes $i \neq m_k$. Thus, we can rewrite transitions replacing the coordinates of $\tilde{z}_t$ by those of $z_t$ at every spot except in the case $i = m_k - 1$ where $\tilde{z}_{t,k,m_k}$ appears through the term $\tilde{z}_{t,k,i+1}$. Transitions are rewritten as follows:

$$f_{t,k,i}(z_t) = \begin{cases} \left[ z_{t,k,i+1} - \left[ d_{t,k} - \sum_{i'=1}^{i} z_{t,k,i'} \right]^+ \right]^+ & \text{for } i = 1, \ldots, m_k - 2, \\ \left[ \tilde{z}_{t,k,m_k} - \left[ d_{t,k} - \sum_{i'=1}^{m_k-1} z_{t,k,i'} \right]^+ \right]^+ & \text{for } i = m_k - 1, \\ z_{t,k,i+1} & \text{for } i = m_k, \ldots, m_k + L_k - 1. \end{cases}$$

$\square$

The case $i \geq m_k$ is trivial, the case $i \in [m_k - 2]$ can be dealt with two applications of Corollary B.5 and for the final case $i = m_k - 1$ we start by differentiating $\delta$ using Proposition B.8 and Example B.6. This leads us to the expressions of $\partial^+ \delta$ and $\partial^- \delta$ and a final application of Corollary B.5 leads to the desired result.

## B.6 CENSORED DEMAND CASE

Here, we assume there are no warehouse-capacity constraints ($V_t = +\infty$) and show that one can compute all the partial derivatives required by GAPSI using censored demand information. In this case, instead of observing the true demands $d_t$, we only observe a sales vector $s_t = (s_{t,k,i})_{k \in [K], i \in [m_k]}$, defined in our model by:

$$s_{t,k,i} = \min \left\{ \tilde{z}_{t,k,i} , \left[ d_{t,k} - \sum_{j=1}^{i-1} \tilde{z}_{t,k,j} \right]^+ \right\}, \tag{11}$$

where $\tilde{z}_t$ is the state-control couple after discarding defined in Equation (4). The policy is completely known when computing its derivatives thus we only consider losses and transitions, which depend on the demand.

Notice that when letting $V_t = +\infty$, there is no overflow, the state-control couple after discarding defined in Equation (4) is given by $\tilde{z}_{t,k,i} = [z_{t,k,i}]^+$ for $i = m_k$ and $\tilde{z}_{t,k,i} = z_{t,k,i}$ otherwise. Since the model is such that state-control couples remain in $\mathbb{R}_+^{n+K}$, we can safely assume $\tilde{z}_{t,k,i} = z_{t,k,i}$ for all $i \in [n_k + 1]$ instead, which will make derivations simpler.

We start by stating a property regarding the sales vector and then proceed to the computation of the left derivatives of losses and transitions in terms of the sales vector.

### B.6.1 PROPERTIES OF THE SALES VECTOR

**Proposition B.11.** *Consider the sales vector $s_t$ defined in Equation* (11)*. Then, we have for all $k \in [K]$, $i \in [m_k]$,*

$$\sum_{j=1}^{i} s_{t,k,j} = \min \left\{ \sum_{j=1}^{i} \tilde{z}_{t,k,j} , d_{t,k} \right\}, \tag{12}$$

*Proof.* This property can be shown by a simple induction of $i = 1, \ldots, m_k$.

For $i = 1$, we clearly have:

$$\sum_{j=1}^{1} s_{t,k,j} = s_{t,k,1} \overset{(11)}{=} \min\left\{\tilde{z}_{t,k,1} , [d_{t,k}]^+\right\} \overset{d_{t,k} \geq 0}{=} \min\left\{\sum_{j=1}^{1} \tilde{z}_{t,k,j} , d_{t,k}\right\}.$$

Assuming Equation (12) holds for some $i \in [m_k - 1]$, we have:

$$\sum_{j=1}^{i+1} s_{t,k,j} \overset{(12)}{=} s_{t,k,i+1} + \min\left\{\sum_{j=1}^{i} \tilde{z}_{t,k,j} , d_{t,k}\right\}$$

$$\overset{(11)}{=} \min\left\{\tilde{z}_{t,k,i+1} , \left[d_{t,k} - \sum_{j=1}^{i} \tilde{z}_{t,k,j}\right]^+\right\} + \min\left\{\sum_{j=1}^{i} \tilde{z}_{t,k,j} , d_{t,k}\right\}.$$

Since the following relations hold:

$$\left[d_{t,k} - \sum_{j=1}^{i} \tilde{z}_{t,k,j}\right]^+ + \min\left\{\sum_{j=1}^{i} \tilde{z}_{t,k,j} , d_{t,k}\right\} = d_{t,k},$$

$$\tilde{z}_{t,k,i+1} + \min\left\{\sum_{j=1}^{i} \tilde{z}_{t,k,j} , d_{t,k}\right\} = \min\left\{\sum_{j=1}^{i+1} \tilde{z}_{t,k,j} , d_{t,k} + z_{t,k,i+1}\right\},$$

we can further simplify the expression of $\sum_{j=1}^{i+1} s_{t,k,j}$,

$$\sum_{j=1}^{i+1} s_{t,k,j} = \min\left\{\sum_{j=1}^{i+1} \tilde{z}_{t,k,j} , d_{t,k} + z_{t,k,i+1} , d_{t,k}\right\} \overset{z_{t,k,i+1} \geq 0}{=} \min\left\{\sum_{j=1}^{i+1} \tilde{z}_{t,k,j} , d_{t,k}\right\}.$$

This concludes the proof. $\qquad\square$

### B.6.2 Losses

**Proposition B.12.** *Consider our model without warehouse-capacity constraints $(V_t = +\infty)$. Then, the loss function defined in Equation (6) can be rewritten as:*

$$\ell_t(z_t) = \sum_{k=1}^{K} \left( c_{t,k}^{\mathrm{purc}} \cdot z_{t,k,m_k+L_k} + c_{t,k}^{\mathrm{hold}} \cdot \left[\sum_{i=1}^{m_k} z_{t,k,i} - d_{t,k}\right]^+ \right.$$

$$\left. + c_{t,k}^{\mathrm{pena}} \cdot \left[d_{t,k} - \sum_{i=1}^{m_k} z_{t,k,i}\right]^+ + c_{t,k}^{\mathrm{outd}} \cdot [z_{t,k,1} - d_{t,k}]^+ \right).$$

*The left partial derivatives of the losses can be written as follows:*

$$\frac{\partial^- \ell_t(z_t)}{\partial z_{t,k',i'}} = c_{t,k'}^{\mathrm{purc}} \mathbb{1}_{\{i'=m_{k'}+L_{k'}\}} + c_{t,k'}^{\mathrm{hold}} \mathbb{1}_{\{i'\in[m_{k'}]\}} \mathbb{1}_{\{\sum_{i=1}^{m_{k'}} z_{t,k',i} > \sum_{i=1}^{m_{k'}} s_{t,k',i}\}}$$

$$- c_{t,k'}^{\mathrm{pena}} \mathbb{1}_{\{i'\in[m_{k'}]\}} \mathbb{1}_{\{\sum_{i=1}^{m_{k'}} z_{t,k',i} = \sum_{i=1}^{m_{k'}} s_{t,k',i}\}} + c_{t,k'}^{\mathrm{outd}} \mathbb{1}_{\{i'=1\}} \mathbb{1}_{\{z_{t,k',1} > s_{t,k',1}\}}.$$

*where $s_t$ is the sales vector defined in Equation (11).*

*Proof.* First, we need to compute the left partial derivatives of the loss functions in the case $V_t = +\infty$. To do so, we can either do it from scratch by applying Corollary B.5 (or Example B.6), or use Proposition B.9 while replacing $\partial^- \tilde{z}_{t,k,m_k}/\partial z_{t,k',i'}$ from Proposition B.8 by $\mathbb{1}_{\{k=k'\}} \mathbb{1}_{\{i'=m_{k'}\}}$.

Either way, we obtain the following left partial derivatives for the losses:

$$\frac{\partial^- \ell_t(z_t)}{\partial z_{t,k',i'}} = c_{t,k'}^{\mathrm{purc}} \mathbb{1}_{\{i'=m_{k'}+L_{k'}\}} + c_{t,k'}^{\mathrm{hold}} \mathbb{1}_{\{i'\in[m_{k'}]\}} \mathbb{1}_{\{\sum_{i=1}^{m_{k'}} z_{t,k',i} > d_{t,k'}\}}$$

$$- c_{t,k'}^{\mathrm{pena}} \mathbb{1}_{\{i'\in[m_{k'}]\}} \mathbb{1}_{\{d_{t,k'} \geq \sum_{i=1}^{m_{k'}} z_{t,k',i}\}} + c_{t,k'}^{\mathrm{outd}} \mathbb{1}_{\{i'=1\}} \mathbb{1}_{\{z_{t,k',1} > d_{t,k'}\}}.$$

Then, using the property (12) from Proposition B.11, we observe that:

$$\sum_{j=1}^{i} z_{t,k,j} > d_{t,k} \iff \sum_{j=1}^{i} z_{t,k,j} > \sum_{j=1}^{i} s_{t,k,j}$$

for all $k \in [K]$, $i \in [m_k]$. Considering this equivalence for $i = m_k$ and $i = 1$ leads to the desired result. □

### B.6.3 TRANSITIONS

**Proposition B.13.** *Consider our model without warehouse-capacity constraints $(V_t = +\infty)$. Then, the transition functions defined in Equation (5) can be rewritten as:*

$$f_{t,k,i}(z_t) = \begin{cases} \left[ z_{t,k,i+1} - \left[ d_{t,k} - \sum_{i'=1}^{i} z_{t,k,i'} \right]^+ \right]^+ & \text{for } i = 1, \dots, m_k - 1, \\ z_{t,k,i+1} & \text{for } i = m_k, \dots, m_k + L_k - 1. \end{cases}$$

*The left partial derivatives of the losses can be written as follows:*

$$\frac{\partial^- f_{t,k,i}(z_t)}{z_{t,k',i'}} = \Big( \mathbb{1}_{\{i \in [n_k] \setminus [m_k - 1]\}} \mathbb{1}_{\{i'=i+1\}} + \mathbb{1}_{\{i \in [m_k-1]\}} \big( \mathbb{1}_{\{i'=i+1\}} \mathbb{1}_{\{f_{t,k,i}(z_t)>0\}}$$

$$+ \mathbb{1}_{\{i' \in [i]\}} \mathbb{1}_{\{\sum_{i''=1}^{i+1} z_{t,k,j} > \sum_{i''=1}^{i+1} s_{t,k,j}\}} \mathbb{1}_{\{\sum_{i''=1}^{i} z_{t,k,j} = \sum_{i''=1}^{i} s_{t,k,j}\}} \big) \Big) \mathbb{1}_{\{k=k'\}}.$$

*where $s_t$ is the sales vector defined in Equation (11).*

*Proof.* As in the proof of Proposition B.12, we first need to compute the left partial derivatives of the transition functions in the case $V_t = +\infty$. To do so, we can either do it from scratch by applying Corollary B.5 twice, or use Proposition B.10 while replacing $\partial^- \tilde{z}_{t,k,m_k} / \partial z_{t,k',i'}$ from Proposition B.8 by $\mathbb{1}_{\{k=k'\}} \mathbb{1}_{\{i'=m_{k'}\}}$.

Either way, we obtain the following left partial derivatives for the transitions:

$$\frac{\partial^- f_{t,k,i}(z_t)}{z_{t,k',i'}} = \mathbb{1}_{\{k=k'\}} \Big( \mathbb{1}_{\{i \in [n_k] \setminus [m_k - 1]\}} \mathbb{1}_{\{i'=i+1\}} + \mathbb{1}_{\{i \in [m_k-1]\}} \big($$

$$\mathbb{1}_{\{i'=i+1\}} \mathbb{1}_{\{z_{t,k,i+1} > [d_{t,k} - \sum_{i''=1}^{i} z_{t,k,i''}]^+\}} + \mathbb{1}_{\{i' \in [i]\}} \mathbb{1}_{\{z_{t,k,i+1} > d_{t,k} - \sum_{i''=1}^{i} z_{t,k,i''} \ge 0\}} \big) \Big).$$

Then, we observe the following equivalences for all $i \in [m_k - 1]$,

$$z_{t,k,i+1} > \left[ d_{t,k} - \sum_{i''=1}^{i} z_{t,k,i''} \right]^+ \iff f_{t,k,i}(z_t) > 0,$$

$$z_{t,k,i+1} > d_{t,k} - \sum_{i''=1}^{i} z_{t,k,i''} \overset{(12)}{\iff} \sum_{i''=1}^{i+1} z_{t,k,i''} > \sum_{i''=1}^{i+1} s_{t,k,i''},$$

$$d_{t,k} - \sum_{i''=1}^{i} z_{t,k,i''} \ge 0 \overset{(12)}{\iff} \sum_{i''=1}^{i} z_{t,k,i''} = \sum_{i''=1}^{i} s_{t,k,i''}.$$

This concludes the proof. □

## C NUMERICAL EXPERIMENTS

In this section, we give additional details on the experiments of the main paper (in Subsections C.1, C.2 and C.3). We also present three new experiments: a simulation with Poisson demands where we use GAPSI in a setup where we know the optimal policy (Subsection C.4), a study on the impact of lifetimes and lead times on the performance (Subsection C.5), and a large scale experiments where we test GAPSI when there are hundreds of products in the inventory (Subsection C.6).

We give the exact definitions of the metrics we computed to evaluate the performances of GAPSI: the *lost sales percentage*, discussed in the main text, and two additional metrics classical in inventory problems, that are, the *outdating percentage* and the *ratio of losses*. They are respectively defined by:

$$100 \times \frac{\sum_{t=1}^{T} \sum_{k=1}^{K} [d_{t,k} - \sum_{i=1}^{m} z_{t,k,i}]^+}{\sum_{t=1}^{T} \sum_{k=1}^{K} d_{t,k}},$$

$$100 \times \frac{\sum_{t=1}^{T} \sum_{k=1}^{K} [x_{t,k,1} - d_{t,k}]^+}{\sum_{t=1}^{T} \sum_{k=1}^{K} u_{t,k}},$$

$$\frac{\sum_{t=1}^{T} \ell_t(x_t, u_t)}{\sum_{t=1}^{T} \ell_t(x_t(S_T^*), u_t(S_T^*))},$$

where $(x_t(S_T^*), u_t(S_T^*))_{t \in [T]}$ is the trajectory associated to $S_T^*$.

For illustration purposes, Figure 6 shows the total sales of the M5 dataset, over the whole horizon and on a specific window.

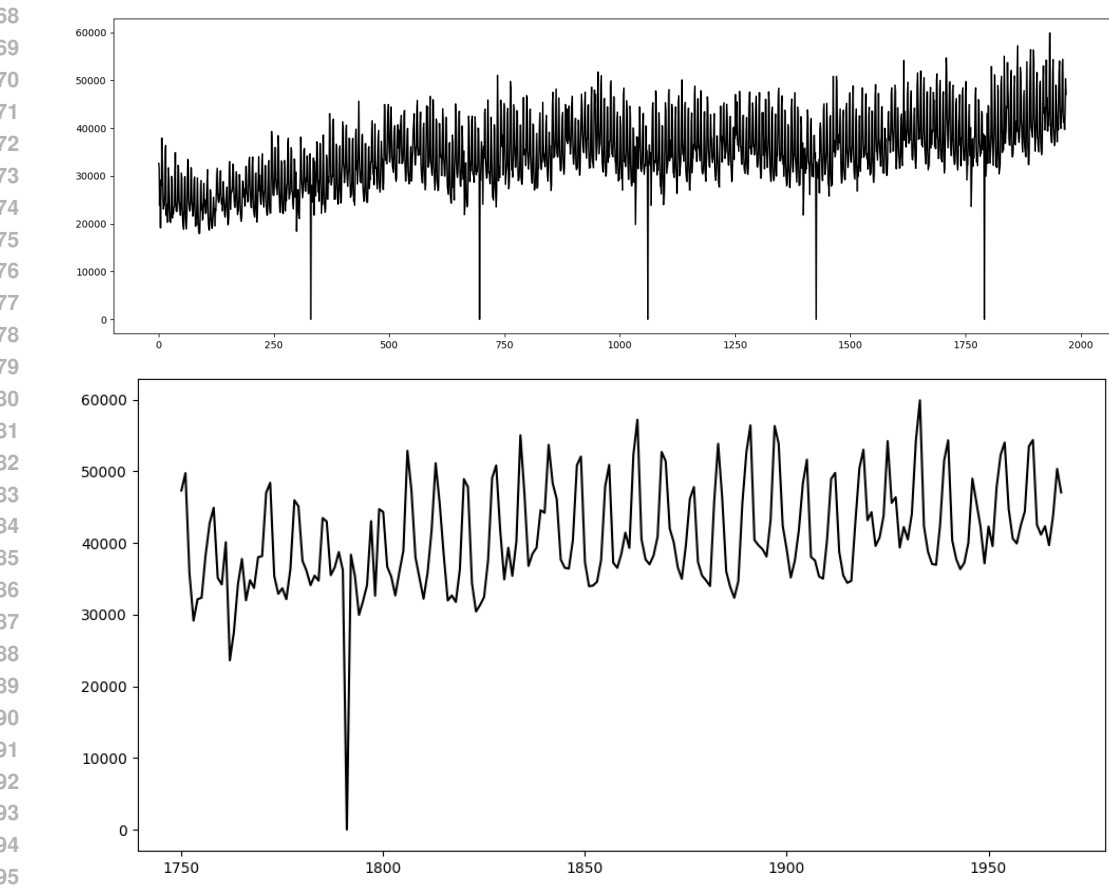

Figure 6: Total demand in the M5 dataset across all the $T = 1969$ time periods (top) and a zoom on the last time periods (bottom).

## C.1 ADDITIONAL RESULTS FOR SECTION 4.1

We give in Table 2 results on the three metrics corresponding to the experiment of Section 4.1. The last column corresponds to the numbers indicated in the legend of Figure 2.

Table 2: Metrics of several algorithms including GAPSI with features, correspondint to Figure 2

|  | Lost sales | Outdating | Ratio of losses |
|---|---|---|---|
| GAPSI without features | 1.02% | 0.09% | 0.952 |
| Best cyclic base-stock policy | 0.75% | 0.08% | 0.906 |
| GAPSI with features | 0.66% | 0.05% | 0.851 |

## C.2 Additional results for Section 4.2

We give in Table 3 the computation time corresponding to the experiment.

Table 3: Computation time in seconds.

|  | M5 | | | Califrais |
|---|---|---|---|---|
|  | Total | Category | Product | Total |
| Base-stock levels $\hat{d}_t$ | 0.22 | 0.22 | 0.21 | 0.08 |
| MPC | 12.38 | 83.98 | 34.80 | 5.93 |
| GAPSI without forecasts | 0.67 | 0.67 | 0.68 | 0.26 |
| GAPSI with forecasts | 0.69 | 0.67 | 0.68 | 0.25 |

## C.3 Additional result for Section 4.3

The normalized standard deviation is defined as:

$$\sqrt{\frac{1}{T} \sum_{t=1}^{T} \left( \frac{d_{t,k}}{\sum_{t'=1}^{T} d_{t',k}/T} - 1 \right)^2}.$$

As an illustration, we show in Figure 7 four examples of demands, corresponding to different levels of variability with respect to this metric. The top curve has lowest normalized standard deviation, while the bottom one has the largest. We can see that for large standard deviation, there are changes of regimes in the demand, with some long periods of zero.

## C.4 Classical perishable inventory systems

In this series of simulations, we evaluate GAPSI as an algorithm for learning stationary base-stock policies in the context of classical perishable inventory systems. More precisely, we consider the experimental setup of Bu et al. (2023, Subsection 7.1). In this setup, there is a single product $K = 1$, which is perishable with a lifetime of $m = 3$ periods and has no order lead time, $L = 0$, and no warehouse-capacity constraint $V_t = +\infty$. Its demand is drawn independently across time periods from a Poisson distribution with mean 5. The losses include purchase costs, holding costs, penalty costs and outdating costs with time-invariant unit costs.

According to the simulations of Bu et al. (2023), in this setup, the best stationary base-stock policy computed with distributional knowledge performs very well with a relative optimality gap of at most $0.48\%$, where the optimal baseline taken into consideration is the following:

$$\text{OPT} = \inf_{\pi \in \Pi} \limsup_{T \to +\infty} \frac{1}{T} \sum_{t=1}^{T} \mathbb{E} \left[ \ell_t(x_t(\pi), u_t(\pi)) \right],$$

with $\Pi = \{(\pi_t)_{t \in \mathbb{N}} \mid \pi_t : \mathbb{X} \to \mathbb{U} \text{ measurable}\}$, that is, $\Pi$ is the set of time-varying policies mapping a state $x_t$ to an order quantity $u_t = \pi_t(x_t)$ through a measurable map $\pi_t$.

The results of the simulations are given in Table 4. Each line of this table corresponds to a set of time-invariant unit costs: $(c^{\text{purc}}, c^{\text{pena}}, c^{\text{outd}})$, and the unit holding cost is fixed to $c^{\text{hold}} = 1$.

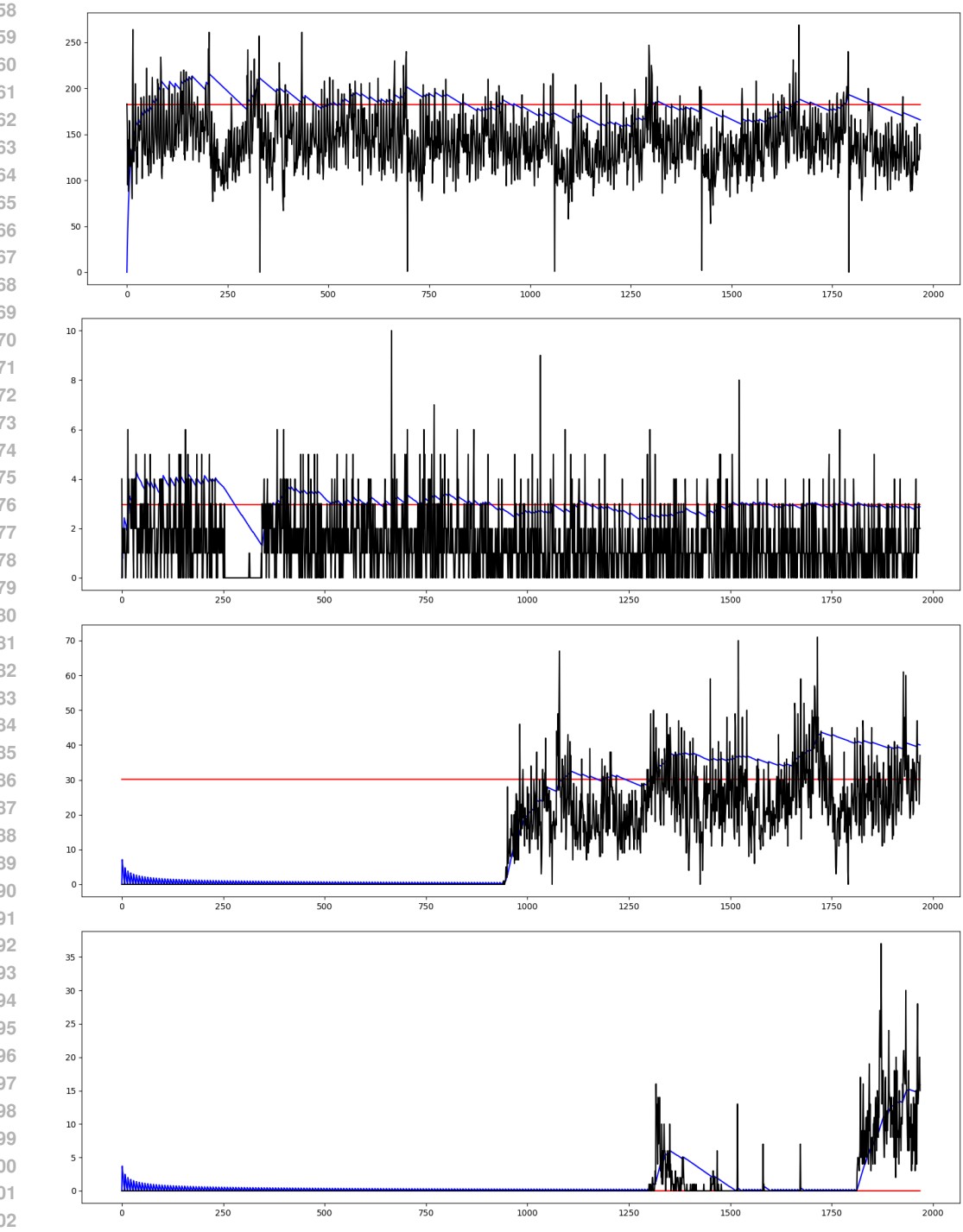

Figure 7: Demands (black), GAPSI's learned base-stock $S_t$ (blue) and best stationary base-stock policy $S_T^*$ (red) in problems with demands' normalized standard deviation of 0.21, 0.96, 1.12 and 3.38 (top to bottom).

Instead of considering the online performances of GAPSI, we measure the performances of base-stock policies with base-stock level learned by GAPSI using the technique of averaging. This is a standard approach when converting an online algorithm to a stochastic optimization algorithm, see online-to-batch conversion (Orabona, 2019, Chapter 3). More specifically, after running GAPSI with $\Theta = [0, 20]$, $w_t = 1$, $\eta = 0.1$ and $B = 10$ against a (training) demand sequence of length

10000, we compute, for each $T \in \{100, 1000, 5000, 10000\}$, the average base-stock level learned $\bar{S}_T = \sum_{t=1}^{T} S_t / T$ and evaluate the expected long-term average loss of the stationary base-stock policy associated to $\bar{S}_T$ using 100 (test) demand sequences of length 10000.

Table 4: Expected average test loss of GAPSI's learned stationary base-stock policies at different points of the learning process and different unit costs.

|            | $T = 100$        | $T = 1000$       | $T = 5000$       | $T = 10000$      | OPT   |
|------------|------------------|------------------|------------------|------------------|-------|
| $(0, 8, 3)$  | $4.32 \pm 0.15$  | $4.22 \pm 0.05$  | $4.20 \pm 0.04$  | $4.19 \pm 0.05$  | 4.16  |
| $(0, 8, 6)$  | $4.41 \pm 0.16$  | $4.27 \pm 0.05$  | $4.27 \pm 0.05$  | $4.26 \pm 0.04$  | 4.23  |
| $(0, 8, 8)$  | $4.45 \pm 0.19$  | $4.31 \pm 0.05$  | $4.31 \pm 0.05$  | $4.31 \pm 0.04$  | 4.28  |
| $(0, 20, 8)$ | $6.01 \pm 0.43$  | $5.58 \pm 0.08$  | $5.58 \pm 0.08$  | $5.57 \pm 0.08$  | 5.50  |
| $(0, 40, 8)$ | $8.12 \pm 1.03$  | $6.61 \pm 0.11$  | $6.62 \pm 0.11$  | $6.62 \pm 0.12$  | 6.56  |
| $(5, 8, 3)$  | $28.16 \pm 0.18$ | $28.03 \pm 0.13$ | $28.03 \pm 0.12$ | $27.99 \pm 0.13$ | 28.01 |
| $(5, 8, 6)$  | $28.18 \pm 0.19$ | $28.05 \pm 0.12$ | $28.04 \pm 0.12$ | $28.02 \pm 0.13$ | 28.02 |
| $(5, 8, 8)$  | $28.16 \pm 0.16$ | $28.04 \pm 0.12$ | $28.02 \pm 0.15$ | $28.04 \pm 0.12$ | 28.03 |
| $(5, 20, 8)$ | $30.79 \pm 0.42$ | $30.30 \pm 0.15$ | $30.30 \pm 0.14$ | $30.30 \pm 0.15$ | 30.26 |
| $(5, 40, 8)$ | $33.12 \pm 0.93$ | $31.65 \pm 0.16$ | $31.64 \pm 0.17$ | $31.63 \pm 0.18$ | 31.57 |

Using Table 4 we can compute the relative optimality gap which is at most $1.25\%$ for $T = 10000$. This indicates that GAPSI can be used to learn almost optimal base-stock policies in classical perishable inventory systems.

### C.5 IMPACT OF THE LIFETIME AND THE LEAD TIME

In this experiment, we study the impact of the lead time and the lifetime on GAPSI's performances in a single-product lost sales FIFO perishable inventory system without warehouse-capacity constraints ($K = 1, V_t = +\infty$). The demand of the product is given by the total demand of the M5 dataset. We consider time-invariant unit costs: $c^{\text{purc}} = 1$, $c_t^{\text{hold}} = 1$, $c_t^{\text{outd}} = 1$ and $c_t^{\text{pena}} = 10$.

For each value of lifetime $m$ and lead time $L$, we ran both the best stationary base-stock policy $S_T^*$ in hindsight of the demand realizations and GAPSI with a single and constant feature: $w_t = (L + 1) \max_{s \in [T]} d_s$, learning rate scale parameter $\eta = 0.1$, buffer size $B = 50$ over $\Theta = [0, 1]$. We chose this constant feature, instead of $w_t = 1$ for instance, so that the parameters can lie in $[0, 1]$ and can be interpreted as a ratio of the maximum demand.

Table 5: Lost sales percentage

|          | $L = 0$ | $L = 1$ | $L = 7$ | $L = 14$ |
|----------|---------|---------|---------|----------|
| $m = 2$  | 1.02%   | 2.45%   | 4.02%   | 5.10%    |
| $m = 7$  | 1.02%   | 2.42%   | 4.94%   | 7.28%    |
| $m = 30$ | 1.02%   | 2.42%   | 4.94%   | 7.20%    |

Table 6: Outdating percentage

|          | $L = 0$ | $L = 1$ | $L = 7$ | $L = 14$ |
|----------|---------|---------|---------|----------|
| $m = 2$  | 0.09%   | 0.26%   | 1.27%   | 4.46%    |
| $m = 7$  | 0%      | 0%      | 0%      | 0.92%    |
| $m = 30$ | 0%      | 0%      | 0%      | 0%       |

The results are given in Tables 5, 6 and 7. We observe that in all scenarios, GAPSI outperforms the best stationary base-stock policy, meaning that GAPSI can achieve *negative* regret. Furthermore,

Table 7: Ratio of losses

|          | $L = 0$ | $L = 1$ | $L = 7$ | $L = 14$ |
|----------|---------|---------|---------|----------|
| $m = 2$  | 0.952   | 0.943   | 0.853   | 0.793    |
| $m = 7$  | 0.954   | 0.951   | 0.825   | 0.958    |
| $m = 30$ | 0.954   | 0.951   | 0.826   | 0.904    |

even in the most difficult settings with high lead time and low lifetime (upper right corner of the tables), GAPSI managed to keep the lost sales moderate: at most 7.28%, and the outdating percentage small: at most 4.46%.

### C.6 LARGE SCALE EXPERIMENTS

Here, we run large scale experiments on the whole M5 dataset at the product level ($K = 3049$) and the proprietary Califrais dataset ($K = 299$) which features perishable products. In the M5 dataset, lifetimes, lead times and costs has been set as usual to $m_k = 3$, $L_k = 0$, $c_{t,k}^{\text{purc}} = 1$, $c_{t,k}^{\text{hold}} = 1$, $c_{t,k}^{\text{outd}} = 1$ and $c_{t,k}^{\text{pena}} = 10$. On the other hand, Califrais dataset already include lifetimes $m_k \in \{3, 4, 5, 6, 7, 10, 30\}$, lead times $L_k \in \{1, 2, 3, 4\}$, time-varying unit purchase costs and time-varying unit selling prices. Purchase costs $c_{t,k}^{\text{purc}}$, holding costs $c_{t,k}^{\text{hold}}$ and outdating cost $c_{t,k}^{\text{outd}}$ has been set to these purchase costs provided and the penalty cost has been set to 10 times the selling prices provided. Let us mention that the Califrais dataset features more erratic demands compared to the M5 dataset and only $T = 860$ days of data. Indeed, the normalized standard deviation in the Califrais dataset lie between 0.82 and 29.33 compared to 0.21 and 3.38 in the M5 dataset. Figure 8 shows two demand sequences from the Califrais dataset. Half of the demand sequences have higher normalized standard deviation than those depicted in this figure.

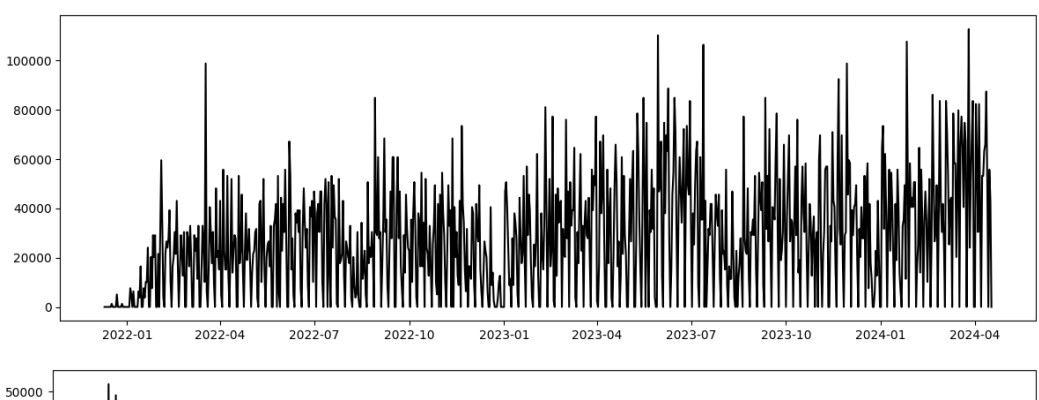

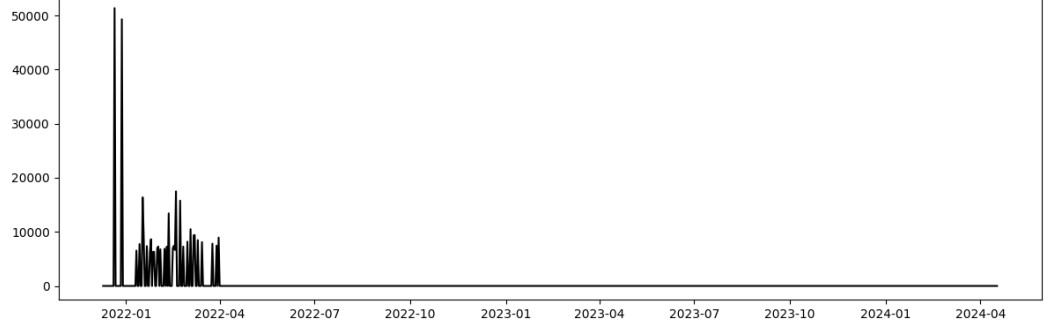

Figure 8: Two demand sequences from the Califrais dataset with respectively lowest normalized std. 0.83 (top) and median normalized std. 6.98 (bottom).

For both datasets, we did not consider volume constraints ($V_t = +\infty$) and GAPSI is run with time-invariant features $w_{t,k} = (L_k + 1) \max_{s \in [T]} d_{s,k}$ and parameters $\eta = 0.1$, $B = 10$ over $\Theta = [0, 1]^K$.

Table 8: Large scale experiments

|  | Lost sales | Outdating |
|---|---|---|
| M5 ($K = 3049$) | 4.86% | 4.74% |
| Califrais ($K = 299$) | 18.05% | 6.07% |

The results are given in Table 8. Overall, the performances are better on the M5 dataset compared to the Califrais dataset even though the former contains more than 10 times the number of products of the latter. This is due to several factors. In Califrais' dataset, the time horizon is shorter ($T = 860$), the replenishment is not instantaneous and the demand is more erratic. As we have seen in Sections 4.3 and C.5, important lead times and variance impact negatively the performances of GAPSI. This experiment shows that the number of products is less an issue compared to the properties of these products (lifetimes, lead times, demands' variance...). We think that employing coordinate-wise learning processes through AdaGrad for hyper-rectangles learning rates is helping dealing with such high-dimensional problems.

