# OpenReview forum: "Online Policy Selection for Inventory Problems"
_ICLR.cc/2025/Conference — Submitted to ICLR 2025_

### Official Review · Reviewer_xtfp · 2024-10-28

**Soundness:** 3
**Presentation:** 2
**Contribution:** 2
**Rating:** 6
**Confidence:** 2

**Summary:**

The paper introduces the GAPSI algorithm, which combines online learning techniques with inventory control theory to address complex inventory management problems. This work notably contributes to practical aspects, especially in dealing with multiple products, perishability, and warehouse capacity constraints.

**Strengths:**

1.  The introduction of the GAPSI algorithm represents a significant advancement in applying online learning techniques to realistic inventory management problems.

2. Addressing Non-Differentiability: The focus on the challenges posed by non-differentiability and the proposed solutions demonstrate a deep understanding of the complexities involved in inventory optimization.

3. Extensive numerical experiments validate the performance of GAPSI.

**Weaknesses:**

1. The language is generally clear, but there are areas where conciseness can be improved. Aim to eliminate redundancy and streamline complex sentences for better readability. Additionally, ensure consistent terminology throughout the paper.

(1) Ensure that terms like "non-differentiability" are consistently used throughout the paper without unnecessary variation. For example, if you introduce "non-differentiability" in one section, use it consistently instead of switching to other similar terms.

(2) Sentences such as "This algorithm is adapted from GAPS (Lin et al., 2024) to take into account specific aspects of inventory problems, in particular, the fact that the functions we are dealing with are not differentiable" can be simplified. You might rephrase it to, "This algorithm adapts GAPS (Lin et al., 2024) to address the non-differentiability of functions in inventory problems."

2. The numerical experiments demonstrate the effectiveness of the proposed algorithm; however, the experimental section could be more rigorous. It is recommended to test under various parameter settings and scenarios to validate the robustness of GAPSI. Additionally, consider incorporating a broader range of performance metrics, in addition to the ratio of cumulative losses, to evaluate the algorithm's effectiveness.

3. The discussion on theoretical guarantees is critical. While the authors mention the unsuitability of standard OPS assumptions for the considered inventory problems, further exploration of how to establish performance guarantees or convergence analysis without these assumptions would be valuable.

**Questions:**

1. What criteria did you use to select the datasets? How representative are these datasets of real-world applications?

2. How do you determine which features to include in the feature-enhanced invarient of this policy? Are there specific criteria or methods for feature selection that you recommend?

3. Can you elaborate on the specific challenges posed by the non-differentiability?

---

> ### Author Response · Authors · 2024-11-18
> **Answer to reviewer xtfp**
>
> ## Weaknesses
>
> 1. Thank you for these readability suggestions, we will make sure to update the final version of the paper accordingly.
>
> 2. More experimental results are available in Appendix C, where we test the robustness of GAPSI with respect to its different parameters (lifetime, lead times, demand variability, number of products). The different subsections of Section 4 in the main paper are also designed to each explore a specific aspect of GAPSI, varying only one parameter (for example, in Subsection 4.1 we explore the impact of the features, in Subsection 4.3 of the variance of the demands etc.).  As for your second point, in terms of metrics, we consider currently the ratio of losses, the lost sales percentage, the outdating percentage and the computational time. Are there any other metrics that you would advise adding? In light of your comment, these aspects of the experiments, and in particular the different metrics used, may not have been emphasized enough, which we will address in the final version. If there are any experiments you would like to see added, we would be happy to include them, as far as possible given the page limit.
>
> 3. We refer to the general answer for the question of obtaining theoretical bounds for GAPSI.
>
> ## Questions
>
> 1. These are real-world datasets, meaning that they correspond to real demands of products from inventories of two companies: Walmart and a food supply chain company. We stress that this is different to many articles in the inventory literature who use only simulated datasets. The Walmart dataset is a very interesting benchmark because it contains five years of historical sales of thousands of products, while the second proprietary dataset is less classical and more challenging (it is more noisy, see Section C.6 in the appendix). If the reviewer knows other classical datasets that we should use we would be happy to include them.
>
> 2. Similar to linear regression problems, feature selection here depends on the problem at hand and typically on the demand process. For instance, if the manager knows that the demand has a seasonal pattern he can incorporate it as features. If, on the other hand, the manager does not possess any information about the demands he can always fall back to classical base-stock policies using constant features.  Moreover, in industry it is common to treat inventory problems in two separate steps: first, develop demand forecasts, then optimize with these forecasts. This approach falls naturally into our method by using these forecasts as features.
>
> 3. We refer to the general answer on the theoretical aspects of GAPSI.

---

> > ### Comment · Reviewer_xtfp · 2024-11-26
> >
> > I appreciate the authors for their detailed explanation. I will update my  score to 6.

---

### Official Review · Reviewer_bqHm · 2024-10-29

**Soundness:** 3
**Presentation:** 3
**Contribution:** 3
**Rating:** 5
**Confidence:** 3

**Summary:**

The article discusses an online inventory management problem where managers must make replenishment decisions based on partial historical information to meet demand and minimize costs. The authors evaluate how realistic, general inventory problems fit within the recent Online Policy Selection (OPS) framework of [1]. Especially, they detail how various constraints very common in the industry, such as
perishability, lead times, or warehouse capacity constraints, can be mathematically modeled in OPS  framework of [1]. Finally, this paper provides extensive experiments which demonstrate the good performance of their GAPSI algorithm, a variant of the GAPS method in [1].

[1] Yiheng Lin, James A Preiss, Emile Anand, Yingying Li, Yisong Yue, and Adam Wierman. On- line adaptive policy selection in time-varying systems: No-regret via contractive perturbations. Advances in Neural Information Processing Systems, 36, 2024.

**Strengths:**

1. The article introduces GAPSI, an algorithm that integrates the GAPS[1] method for the inventory control problems, providing a sophisticated framework to tackle the real-world inventory management.

2. The author explain how common industry constraints, including perishability, lead times, and warehouse capacity limitations, can be effectively formulated as mathematical models within GAPS[1] framework in section 2.2.

3. The algorithm's performance is rigorously tested through extensive numerical simulations using real-world data, which strengthens the credibility of the results and the algorithm's practical applicability.

**Weaknesses:**

1. After meticulously reading this article, I feel that the biggest issue of this paper is that it merely provides a detailed implementation guide (such as policy settings, loss function selection, and modeling details) and an empirical simulation evaluation of  how the GAPS in article [1] can be applied to real inventory replenishment management.

2. Due to a series of problems like 'policies, losses, and transitions are not differentiable, thus neither classical chain rules nor smoothness apply,' the authors have not provided a regret analysis. Although this problem is extremely challenging, a rigorous theoretical analysis is something that everyone hopes to see and have addressed for many researchers in the operations research community and computer theory.

**Questions:**

This article is primarily inclined towards empirical evaluation and presenting some implementation details regarding how the GAPS framework in article [1] can be applied to real inventory replenishment management. Therefore, I would like the authors to demonstrate the significance of these empirical evaluations？ Is it merely to draw our attention to this issue? If that's the case, I feel that this article is more like a heuristic guidance manual for inventory management that could be posted on arXiv or SSRN.

My concern is that this article resembles a case study, lacking comprehensive theoretical proof and large-scale real-world practice, so I would only give it a score of 5.

---

> ### Author Response · Authors · 2024-11-18
> **Answer to reviewer bqHm**
>
> ## Weaknesses
>
> 1. Our paper is indeed methodological, however the proposed algorithm, GAPSI, is not a straightforward application of GAPS [1]. Indeed, inventory problems do not satisfy the assumptions of [1], which has led to practical issues like stagnation behaviors solved using custom derivative selection. In addition, we provided a new parametrized policy that allows managers to incorporate their knowledge and different learning rates which are adapted to large-scale situations. The setting we propose is very general and allows for many classical settings from operational research.
>
> 2. We refer to the general answer for the question of obtaining theoretical bounds for GAPSI.
>
> ## Questions
>
> Our experiments demonstrate the very good performance of GAPSI on real-world data (from Walmart and a food supply chain company) and against various competitors. Our experiments (Section 4.2) show that GAPSI performs better than several competitors including Model Predictive Control. We will update the experiments section in the final version to highlight these facts, which we understand were not clear. Online Policy Selection is a new framework that is largely unknown to the inventory management community, which explains this performance gap. We believe that our contributions on the empirical side are therefore twofold: (1) we provide a competitive algorithm to the operational research community, (2) we bring these applications to the attention of the machine learning community, which raises new practical and theoretical questions.

---

### Official Review · Reviewer_9oSM · 2024-11-01

**Soundness:** 3
**Presentation:** 4
**Contribution:** 2
**Rating:** 5
**Confidence:** 3

**Summary:**

The paper studies online policy selection for general finite-horizon inventory problems, and generalizes Lin et al. (2024) to develop a new online control algorithm, named GAPSI. Two real-world datasets are used to empirically validate the effectiveness of GAPSI.

**Strengths:**

The paper is very well-written, and the authors' approach is easy to follow. The authors successfully apply online policy selection into inventory control, where the loss function, policy selection function and transition function are all nonsmooth. The authors explain in detail why they need to adopt customized differentiation instead of auto differentiation to make the online algorithm work.

**Weaknesses:**

The methodological contribution is a bit limited to the ML community. For more details, see the questions below.

**Questions:**

The major contribution is that you identified a customized way for the Jacobian computation for inventory problems. Yet, the current treatment seems to be very specialized to inventory problems. Can you generalize your approach to more general nonsmooth control problems by applying more general smoothing techniques (e.g., [1])?

No theoretical regret bounds have been provided. This is mainly because the inventory problem violates some of the assumptions needed for the analysis of GAPS [2]. However, there are other papers that provides theoretical bounds on policy gradient methods for inventory control, see, e.g. [3][4], and thus I believe some similar bounds could also be obtained for GAPSI. Can you thus provide some theoretical bounds for GAPSI?

[1] Nesterov, Y. (2005). Smooth minimization of non-smooth functions. Mathematical programming, 103, 127-152.

[2] Lin, Y., Preiss, J. A., Anand, E., Li, Y., Yue, Y., & Wierman, A. (2024). Online adaptive policy selection in time-varying systems: No-regret via contractive perturbations. Advances in Neural Information Processing Systems, 36.

[3] Bhandari, J., & Russo, D. (2024). Global optimality guarantees for policy gradient methods. Operations Research.
[4] Chen, X., Hu, Y., & Zhao, M. (2024). Landscape of Policy Optimization for Finite Horizon MDPs with General State and Action. arXiv preprint arXiv:2409.17138.

---

> ### Author Response · Authors · 2024-11-18
> **Answer to reviewer 9oSM**
>
> ## Weaknesses
>
> **Answer** This paper indeed focuses on inventory problems, which are not the primary interest of the ML community but are still very important in several industries. Our goal in submitting this article to a ML audience is threefold:  (1) we frame online inventory problems in a mathematical setting where tools from machine learning can be applied (precisely, online policy selection), opening up new applications  (2) we provide a flexible and competitive algorithm (3) we draw the attention of the machine learning community to new theoretical questions. We realized that this last point was not explicit and detailed enough, and we will take care to update it in the final version of the paper, as discussed more thoroughly below.
>
> ## Questions
>
> *« The major contribution is that you identified a customized way for the Jacobian computation for inventory problems. Yet, the current treatment seems to be very specialized to inventory problems. Can you generalize your approach to more general nonsmooth control problems by applying more general smoothing techniques (e.g., [1])? »*
>
> **Answer 1:**
> We refer the reviewer to our general answer on theoretical aspects and stress that obtaining a general approach for non-smooth control problems is a hard and open question.
>
> *« No theoretical regret bounds have been provided. [...] Can you thus provide some theoretical bounds for GAPSI? »*
>
> **Answer 2:**
> We refer to the general answer for the question of obtaining theoretical bounds for GAPSI. Thank you for the references concerning policy gradient methods. This is an interesting direction, however policy gradient methods and GAPS are very different algorithms.
> First, the former operate in the context of markov decision processes where we make a stochasticity assumption (as in [3] and [4]), whereas GAPS solves problems which are *non-stochastic* online control problems. Second, although both methods are first-order methods, they aim at minimizing different functions: policy gradient methods consider the total expected cost whereas GAPS consider the total incurred cost. Due to these important differences, it is far from trivial to adapt these methods to our setting under our realistic assumptions.

---

> > ### Comment · Reviewer_9oSM · 2024-11-26
> >
> > I thank the authors for the explanations. I do think the major weaknesses persist, so I will keep my score.

---

### Official Review · Reviewer_ULHb · 2024-11-01

**Soundness:** 3
**Presentation:** 4
**Contribution:** 2
**Rating:** 3
**Confidence:** 5

**Summary:**

The authors address the challenge of inventory control in dynamic, real-world scenarios, where managers must make sequential replenishment decisions based on past data rather than assuming known demand distributions. It introduces an algorithm built on the Online Policy Selection framework. The approach leverages adaptive learning rates and feature-enhanced base-stock policies to model and manage inventory systems more flexibly. The paper demonstrates the potential of online learning methods for realistic inventory optimization.

**Strengths:**

The approach of applying OPS on inventory problems is creative. The idea of linear base stock policies is a nice concept despite being used in many other applications.

Observations about non-differentiable points are very informative and nicely show why this aspect should not be neglected.

**Weaknesses:**

A lot of the material in the main body (and the appendix) is standard and should be omitted. The entire Section 2.2 is well known in the operations management and operations research communities.

Consideration of base-stock policies is questionable. Safety stock (s,S) policies are much more common in practice and nicely capture fixed costs.

I read with enthusiasm the arguments and illustrations why non-differentiable points are important. However how to copy with this issue left me empty handed. The solution proposed is very simple and begs for additional exploration (for example, in reLU at zero any number in [0,1] is a gradient) and thus selecting the left gradient might not be 'optimal.'

Perhaps the weakest point is that the weights in the parametric policy must be determine by hand. This is a very tall order and a very questionable requirement.

Conducting abundant research in inventory management in my early career, the paper uses many established principles and concepts. On the optimization side, there are very few novel ideas (that really don't rely on aspects of inventory management).

The appendix is a dumpster of many topics, most of them known by prior works.

**Questions:**

1. Why not using safety-stock policies?
2 Are there any guidances, ideally scientific approaches, on how to set up weights?
3. The OR community has developed many heuristics. Why did you benchmark only against one?

---

> ### Author Response · Authors · 2024-11-18
> **Answer to reviewer ULHb**
>
> ## Weaknesses
>
> *« A lot of the material in the main body (and the appendix) is standard and should be omitted. [...] »*
>
> **Answer 1**
> The material you mention is standard in the operational research community but not particularly well known to a machine learning audience. Therefore, we feel that this content should not be omitted entirely. In the final version, we will move some of the material from Section 2.2 to the appendix, so that the averted reader gets a smoother read but the material remains accessible to a wider audience.
>
> *« Consideration of base-stock policies is questionable. [...] »*
>
> **Answer 2**
> Considering fixed costs or (s,S) policies introduces additional difficulty in the problem, in particular discontinuities which are even more challenging than the non-differentiability discussed in Section 3.3. Given that the analysis with base-stock policies is already involved, we feel that this additional complexity is not desirable. To our knowledge, base-stock policies are very common in practice, do you have any reference that would say otherwise?
>
> *« I read with enthusiasm the arguments and illustrations why non-differentiable points are important. [...] »*
>
> **Answer 3**
> Thank you for your interest in this point. The solution we propose is to choose the ‘good’ derivatives, that are the ones that avoid stagnation at zero, for a specific choice of transition functions, losses, and policies. On the one hand, many real-world perishable inventory problems are particular cases of our model. In this case, the formulae given in Appendix B.5 can be directly applied to compute the gradients. On the other hand, for other inventory problems, the strategy we have used can be applied similarly by computing the gradients. We will add some guidelines in the final version on this aspect, so that practitioners in inventory problems do not feel empty handed. Moreover, our goal is to raise awareness on this issue when most of the machine learning community working on gradient-based optimization uses auto-differentiation. It is rare that auto-differentiation causes problems such as the ones we encounter and we thought it interesting to discuss it, since it is a major issue for applying powerful tools such as GAPS to inventory problems.
>
> *« Perhaps the weakest point is that the weights in the parametric policy must be determined by hand. [...]»*
>
> **Answer 4**
> We think that, on the contrary, this is one of the strengths of our feature-enhanced base-stock policy. The manager that has no prior knowledge on the demand process can always fall back to classical base-stock policies by considering constant features. On the other hand, the informed manager that knows for instance the seasonality of the demand can drastically improve the performances of the algorithms by considering seasonality features. Moreover, in industry it is common to treat inventory problems in two separate steps: first, develop demand forecasts, then optimize with these forecasts. This approach falls naturally into our method by using these forecasts as features.
>
> *« Conducting abundant research in inventory management in my early career, the paper uses many established principles and concepts. [...] »*
>
> **Answer 5**
> We are fully aware that we are using classical concepts from an operational research perspective. The novelty of our work lies not in this, but in the link we make between these classical concepts and online policy selection. In applying OPS techniques to these standard problems, we run into both theoretical and practical problems. On the practical side, we encounter the problem of non-differentiability, which is entirely dependent on the specific aspects of inventory management. On the theoretical side, the classical assumptions for studying GAPS are not verified. In the final version we will make sure to clarify these points and we plan to add a subsection dedicated to theoretical aspects, showing how classical assumptions in OPS are not verified for inventory problems.
>
> ## Questions
>
> 1. See Answers 2 and 4 above.
>
> 2. To our knowledge, the literature on these problems is divided into two main approaches. On the one hand, parametric inventory policies such as base-stock policies, and, on the other hand, general control approaches such as Model Predictive Control. In the experiments in the main paper and the appendix, we benchmark against both approaches, more precisely the naive base-stock policy with $S_t = \hat{d}_t$, MPC, the optimal stationary base-stock policy and the optimal cyclical base-stock policy. Can the reviewer point us to the benchmark they think we should compare to? We stress that in this paper we compare methods which are able to handle problems as general as our framework, in particular involving perishability constraints and lead times.

---

### Author Response · Authors · 2024-11-18
**General answer to all reviewers**

We thank all reviewers for their time and interest in our work. We appreciate the positive feedback and address the various weaknesses each reviewer pointed out below. Several reviewers asked about theoretical guarantees, we therefore make a general answer on this point.

We agree that theoretical guarantees would be a great addition to the article. However, we are dealing with non-differentiable control problems which are very hard to deal with.

First, it is well-known that non-differentiability makes dynamical systems very challenging to analyze and control (see Section 4.3 of [1]). Most theoretical results are obtained in online control theories with linear dynamics. On this aspect, the recent works of Lin et al. [3] does already a great leap forward by replacing these linear dynamics by smooth ones. However the classical dynamics in inventory problems are not smooth and developing a general theory of non-smooth online control is an open question.

Second, even focusing on particular inventory problems is theoretically far from trivial. For example, recent works developed in [2] showed that the class of single-product lost sales inventory problems with instantaneous replenishment can be proven to be statistically hard (that is, there is a linear regret lower bound) when the demands are arbitrary (see Proposition 13 of [2]). This is why most of the literature on online inventory problems that obtain theoretical guarantees have to deal with specific problems and assumptions (such as i.i.d. demands) and solve each of them with a different algorithm.

Given how complicated it is to obtain theoretical guarantees for such problems and the fact that our main objective in this paper is to provide an algorithm that can be adapted to a wide range of real-world inventory problems and has strong empirical performances, we have decided not to include theoretical aspects in our initial submission. However, given that this remark was made by several reviewers, we propose to add a dedicated section regarding theoretical guarantees for the revised version.

[1] Blondel, V. D., & Tsitsiklis, J. N. (2000). A survey of computational complexity results in systems and control. Automatica, 36(9), 1249-1274.

[2] Hihat, M., Gaïffas, S., Garrigos, G., & Bussy, S. (2024). Online inventory problems: beyond the iid setting with online convex optimization. Advances in Neural Information Processing Systems, 36.

[3] Lin, Y., Preiss, J. A., Anand, E., Li, Y., Yue, Y., & Wierman, A. (2024). Online adaptive policy selection in time-varying systems: No-regret via contractive perturbations. Advances in Neural Information Processing Systems, 36.

---

### Meta-Review · Area_Chair_TC1G · 2024-12-21

**Metareview:**

There is a consensus among the reviewers that the paper is more suitable to be a implementation manual than a novel academic research paper. I can see its practical value to a field practitioner in the inventory control field, helping his/her day-to-day work (and leveraging their existing knowledge and field insight). But unfortunately, it falls short of the technical rigor and novelty needed for ICLR. I must therefore reject the paper.

**Additional Comments On Reviewer Discussion:**

NA

---

### Decision · Program_Chairs · 2025-01-22

Reject